# Understanding Neural Coding on Latent Manifolds by Sharing Features and Dividing Ensembles

**Martin Bjerke[1], Lukas Schott[2], Kristopher T. Jensen[3],**
**Claudia Battistin[1], David A. Klindt[*, 1], Benjamin A. Dunn[*, 1]**
[1]Norwegian University of Science and Technology, [2]Bosch Center for Artificial Intelligence,
[3]University of Cambridge, [*]Equal contribution
{martin.bjerke,benjamin.dunn}@ntnu.no, klindt.david@gmail.com

## Abstract

Systems neuroscience relies on two complementary views of neural data, characterized by single neuron tuning curves and analysis of population activity. These two perspectives combine elegantly in neural latent variable models that constrain the relationship between latent variables and neural activity, modeled by simple tuning curve functions. This has recently been demonstrated using Gaussian processes, with applications to realistic and topologically relevant latent manifolds. Those and previous models, however, missed crucial shared coding properties of neural populations. We propose *feature sharing* across neural tuning curves which significantly improves performance and helps optimization. We also propose a solution to the *ensemble detection* problem, where different groups of neurons, i.e., ensembles, can be modulated by different latent manifolds. Achieved through a soft clustering of neurons during training, this allows for the separation of mixed neural populations in an unsupervised manner. These innovations lead to more interpretable models of neural population activity that train well and perform better even on mixtures of complex latent manifolds. Finally, we apply our method on a recently published grid cell dataset, and recover distinct ensembles, infer toroidal latents and predict neural tuning curves in a single integrated modeling framework.

## 1 Introduction

Neural population activity can appear high-dimensional (Stringer et al., 2019), yet much recent work has reported that neural populations in higher brain areas are often confined to low dimensional subspaces (Yu et al., 2008; Harvey et al., 2012; Mante et al., 2013; Stokes et al., 2013; Shenoy et al., 2013; Kaufman et al., 2014; Sadtler et al., 2014; Gallego et al., 2017; Elsayed & Cunningham, 2017; Gao et al., 2017). The bread and butter of classic systems neuroscience is linking neural activity to experimentally controlled or observable covariates such as orientation (Hubel & Wiesel, 1979), pitch (Lewicki, 2002), movement (Churchland et al., 2012; Kao et al., 2015), posture (Mimica et al., 2018) and orientation in space (Taube et al., 1990). These two parallel streams of neuroscientific research might at first seem to be at odds with each other (Kriegeskorte & Wei, 2021); tuning studies of individual neurons give a very different picture of neural coding than distributed representations over high-dimensional neural populations. However, they combine elegantly in the form of (neural) latent variable models (LVMs, see Lawrence, 2003; Yu et al., 2008; Pandarinath et al., 2018).

In their basic form, neural LVMs find the low-dimensional structure of neural population activity, for instance, when a large network of neurons is coding mostly along few linear subspaces (Mante et al., 2013; Gao et al., 2017). One advantage is that these models can help us discover latent variables which may not be tracked as classical covariates in systems neuroscience. However, when the mapping from latent variables to predicted spike rate (decoding) is fully unconstrained, e.g., by using a multi-layer neural network, we lose the simple biological interpretation of tuning curves.

In an effort to maintain a biologically interpretable relationship between the latent variables and the neural activity, recent work has proposed more constrained decoders approximating simple tuning

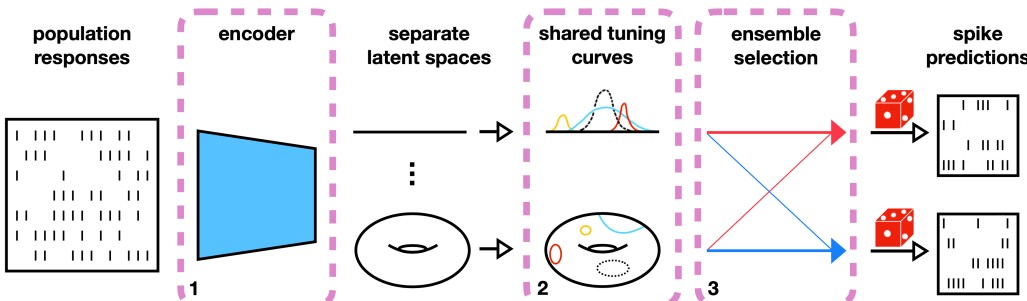

Figure 1: **Model outline**. Our main contributions are outlined in purple. The input is a matrix of neural spiking activity. The encoder (**1**) is a multilayered neural network. The latent spaces are separate with, potentially, different topologies (e.g. $\mathbb{R}^1$ and $\mathbb{T}^2$). The decoder is a parametric tuning curve model with feature sharing (**2**). The ensemble detection is a weighted (ideally one-hot) selection of latent spaces for each neuron (**3**). For decoding of the activity, we assume Poisson spiking.

curves. These tuning curves have been parameterized as Gaussian processes in the framework of Gaussian process latent variable models (GPLVM, Wu et al., 2017; 2018). Through the tuning curve approach, we limit ourselves to biologically plausible solutions that reveal the actual algorithmic structure of the neural system. Thus, LVMs with simple tuning curve decoders bring together the view on neural populations as distributed representations of low-dimensional latent variables, along with the biologically meaningful perspective on individual neural tuning properties.

Some neural populations exhibit topologically interesting latent manifolds (Singh et al., 2008; Peyrache et al., 2015a; Gardner et al., 2022). For instance, grid cells represent navigational space in toroidal coordinates of spatially repeating two dimensional hexagonal grids (Hafting et al., 2005). They appear in different ensembles, commonly referred to as modules, each coding for space at different resolutions (Fyhn et al., 2007; Stensola et al., 2012). Thus, the complete population of grid cells is best described as a collection of ensembles of neurons, where neurons in each ensemble have tuning curves of specific shapes on their respective toroidal latent representations of space (Curto, 2017). By contrast, a two-dimensional Euclidean representation might also account for the complete population of grid cells, but would also completely obscure their efficient and theoretically interesting coding scheme for representing space (Solstad et al., 2006; Sreenivasan & Fiete, 2011; Mathis et al., 2012; Wei et al., 2015; Klukas et al., 2020). A driving motivation behind this work was to model this beautiful neural structure with an LVM that separates the algorithmic and biological parts, while uniting shared tuning properties to be more accurate and trainable than previous approaches.

We propose to train neural LVMs that not only have simple tuning curve decoders, but are also fully differentiable. Thus, we use a flexible encoder, i.e., a neural network as in variational autoencoders (Kingma & Welling, 2014; Rezende et al., 2014), and a simple tuning curve based decoder, akin to GPLVM. The encoder can readily be made convolutional to allow for better latent estimation from adjacent time points. Additionally, we implement a feature basis for the tuning curve shapes in the decoder which can be shared across neurons. We demonstrate that the neural feature sharing, along with the variational end-to-end training, vastly improves both the training stability as well as the final performance of neural LVMs. Moreover, we propose hybrid inference at test-time and show that this, again, brings a considerable improvement in performance. Finally, we integrate the problem of separating distinct ensembles of neurons into our approach — a crucial task for the discovery of different biological structures and the precise mathematical understanding of their topological tuning properties. An illustration of our approach is provided in Fig. 1.

To summarize, our full model performs the task of finding latent variables, separating distinct ensembles of neurons and fitting the prototypical tuning curves on each ensemble's latent space in a single efficient framework. In the following, we therefore refer to our model as **f**eature sharing **a**nd **e**nsemble detection **L**atent **V**ariable **M**odel or `faeLVM`.

## 2 BACKGROUND

Let $\lambda_i$ be the instantaneous firing rate of a neuron $i$. To relate this to the spiking activity $x_i$, we assume a Poisson noise model $x_i \sim \mathcal{P}(\lambda_i)$. We define latent variables $z := \{z_1, \ldots, z_k\}$ (in distinct

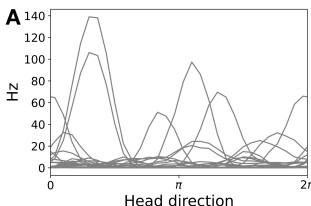 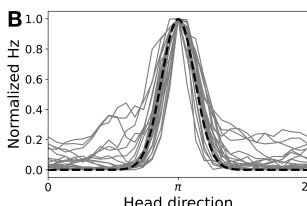

Figure 2: **The feature sharing assumption**. **A**, Tuning curves for 26 head direction tuned neurons. **B**, The same 26 tuning curves, shifted and scaled to visualize structural similarity, along with a standard Gaussian shape (dotted line).

spaces and possibly with different topologies) that change over time and to which neurons are tuned. More precisely, we suppose that there exists a deterministic function $f_i$, which for every neuron $i$ relates the latent variables to the firing rate, i.e., $\lambda_i = f_i(z)$. One additional assumption is that each $f_i$ is only *sensitive* to a subset $J_i$ of latent variables and *invariant* to all others. That means, for two values $z, \hat{z}$ with $z_j = \hat{z}_j \, \forall j \in J_i$ and $z_j \neq \hat{z}_j \, \forall j \notin J_i$ we have $f_i(z) = f_i(\hat{z})$.

As a concrete example, consider a mixed population of head direction neurons $I_1$ and a grid cell module $I_2$. The former are tuned to a circular latent variable $z_1 \in \mathbb{S}^1$ (Peyrache et al., 2015a; Rybakken et al., 2019; Rubin et al., 2019; Chaudhuri et al., 2019), the latter to a toroidal latent variable $z_2 \in \mathbb{T}^2$ (McNaughton et al., 2006; Fuhs & Touretzky, 2006; Gardner et al., 2022). That means, for every neuron in the head direction *ensemble* we have $J_i = \{1\}, \forall i \in I_1$; whereas for every neuron in the grid cell *ensemble* we have $J_i = \{2\}, \forall i \in I_2$. Conceivably, there exist shared latent variables, e.g. attentional state, running speed or pupil dilation, to which multiple or even all neurons are tuned.

Within each ensemble, the tuning functions $f_i$, $i \in I_j$ are likely to have similar structure (Albright, 1984; Taube et al., 1990; Hafting et al., 2005). This is the concept of *feature sharing* (Klindt et al., 2017), which in visual systems neuroscience has proven to be a useful assumption (Batty et al., 2017; Sinz et al., 2018; Ecker et al., 2018; Walker et al., 2018; Cadena et al., 2019; Ustyuzhaninov et al., 2019; Cotton et al., 2020; Zhuang et al., 2021; Burg et al., 2021; Bashiri et al., 2021; Safarani et al., 2021; Franke et al., 2021; Lurz et al., 2021; Nayebi et al., 2021; Seeliger et al., 2021; Goldin et al., 2021; Ustyuzhaninov et al., 2022). In vision, this can be motivated by the fact that visual (especially early retinal) neurons form mosaics, that tile the input space and compute the same response function (i.e., tuning curve) across space (Wässle & Riemann, 1978). Precisely, let $f_\mu$ denote the response function of a visual neuron with receptive field center $\mu \in \mathbb{R}^2$, let $T : \mathbb{R}^2 \to \mathbb{R}^2$ be a spatial translation, and let $z : \mathbb{R}^2 \to \mathbb{R}^1$ be an image (with slight abuse of notation, $z$ is, here, a function that assigns a grey scale value to every position in two-dimensional space, i.e., an image). Let us think of $f_i = f_\mu$ and $f_{i'} = f_{T(\mu)}$ as two neurons of the same type (i.e., same tuning curve shape) but with their receptive field centers at different locations $\mu$ and $T(\mu)$. Then we have $f_i(z) = f_{i'}(z \circ T)$. Thus, for visual neurons we assume *translational equivariance*.

Consequently, we should be able to marginalize over spatial translations to learn the shared structure among all $f_i$ as demonstrated in Klindt et al. (2017). The same intuition holds for populations of auditory neurons that are equivariant to pitch, i.e., that have similar tuning properties translated across log-frequency space (Kell et al., 2018; Kell & McDermott, 2019) and possibly other sensory neurons such as somatosensory populations (Lieber & Bensmaia, 2019). But it also holds true for higher cognitive neurons, such as the ones discussed above, where, for instance, each grid cell is tuned to a different location on the torus and the receptive fields (tuning curves) across grid cells are remarkably similar in shape (Hafting et al., 2005; Fyhn et al., 2007; Gardner et al., 2022). Thus, we propose to introduce *feature sharing* also in this case. More precisely, we argue that every $f_i$ within an ensemble $I_j$, should be modeled by a shared tuning curve function $g_j(z, \theta_i) = f_i(z)$ that applies a simple neuron specific transformation, parameterized by $\theta_i$, to obtain the specific $f_i$ of each neuron. For instance, in the case of a neuron-specific spatial translation ($\mu_i$, the receptive field center as above) and scaling ($\alpha_i$), we would define $\theta_i = \{\mu_i, \alpha_i\}$ and thus $g_j(z, \theta_i) = \alpha_i g_j(T_{\mu_i}(z), \theta_0) = f_i(z)$ ($\theta_0$ yielding simply the centered prototype of the shared tuning curve). Evidence for this assumption is provided in Fig. 2, where centering and rescaling all tuning curves of head direction selective neurons on $\mathbb{S}^1$ (recorded by Peyrache et al., 2015b) clearly exhibits a shared, Gaussian-like tuning.

In visual perception there is translational equivariance, however, only among cells of the same types (Wässle & Riemann, 1978). Therefore, previous work proposed feature sharing in combination with functional cell type identification (Klindt et al., 2017). Specifically, equating neural ensembles with cell types in the retina, this posits that all neurons in a given ensemble $I_j$ be tuned to the same set of latent variables $J_j$, i.e., $J_i = J_j, \forall i \in I_j$. As an example, the toroidal structures $z_j \in \mathbb{T}^2$ of different

grid cell ensembles $I_j$ only become apparent after the successful separation of distinct modules with different spatial resolutions (Gardner et al., 2022). Thus, we argue that feature sharing hinges on the successful identification and separation of distinct neural ensembles in a mixed population recording.

The problem of *ensemble detection* can be formalized as partitioning $n$ recorded neurons $I = \bigcup_j^k I_j$ into $k$ non-empty subsets $I_j \cap I_{j'} = \emptyset, \forall j \neq j'$. Unfortunately, this is a combinatorial problem with $k^n$ possible states. Even for a 'simple' problem such as clustering neurons into $k = 2$ groups, this already exceeds the number of atoms in the universe ($10^{82}$) at $n \geq 273$. With modern neuroscience routinely yielding thousands of recorded neurons per experiment (Jun et al., 2017), brute-force exhaustive search is clearly not a viable approach. To mitigate this issue, we propose a soft-relaxation of the clustering problem. Specifically, we define a convex combination of weights over responses derived from tuning curves on each latent variable, and train the neurons to maximize their likelihood given those weights in a differentiable fashion (explained in more detail in the next section).

## 3 METHODS

As above, we denote neural activity $x$ and the collections of latent variables as $z$. We want to learn a latent variable model $p(x, z) = p(x|z)p(z)$. We choose a variational autoencoder (VAE, Kingma & Welling, 2014), which maximizes a lower bound to the data likelihood called the evidence lower bound, which necessitates the introduction of an approximation to the posterior distribution, i.e., a variational (input dependent) posterior with non-zero support across the domain, $q(z|x) > 0, \forall z, x$.

At test-time, we focus on a precise estimate for the latents $z$. These latents are usually inferred with the encoder that computes the approximate variational posterior $q(z|x)$. Here, we aim to further refine this estimate by performing inference based on the decoder $p(x|z)$. Technically, this can be achieved by directly performing gradient descent on the latents, with the decoder starting from the encoder estimate (Schott et al., 2018; Ghosh et al., 2019) (see Appx. J for further details).

For Euclidean latent spaces $z \in \mathbb{R}^n$, we can simply rely on the standard VAE framework, using a Normal ($\mathcal{N}$) posterior and prior for those dimensions. For spherical latent spaces $\mathbb{S}^n$, we use a wrapped Normal distribution ($w\mathcal{N}$), akin to previous work (Falorsi et al., 2019; Jensen et al., 2020), which satisfies the topology of the latent space. For the relevant case of a toroidal latent space, we can use $\mathbb{T}^2 = \mathbb{S}^1 \times \mathbb{S}^1$, together with the fact that the variational posterior is, usually, assumed to factorize across dimensions. This factorization also helps keep the equations the same when working with multiple latents (corresponding to multiple ensembles), i.e., $z = \{z_1, ..., z_k\}$.

More precisely, we assume that each latent $z_j$ lives either in Euclidean space $\mathbb{R}^{n_j}$ or on a torus $\mathbb{T}^{n_j} = \mathbb{S}^1_1 \times \ldots \times \mathbb{S}^1_{n_j}$ (with $\mathbb{S}^1 = \mathbb{T}^1$). Further, we let our prior factorize (Kingma & Welling, 2014), i.e.,

$$p(z) = \prod_{j=1}^k p(z_j), \quad p(z_j) = \prod_{l=1}^{n_j} p(z_j^{(l)}), \quad z_j^{(l)} \sim \left\{ \begin{array}{l} \mathcal{N}(0, 1), \text{ if } z_j \in \mathbb{R}^{n_j} \\ w\mathcal{N}(0, 1), \text{ if } z_j \in \mathbb{T}^{n_j} \end{array} \right., \quad (1)$$

giving uniform prior distributions. Analogously, we let our variational posteriors factorize as

$$q(z|x) = \prod_{j=1}^k q(z_j|x), \quad q(z_j|x) = \prod_{l=1}^{n_j} q(z_j^{(l)}|x), \quad z_j^{(l)} \sim \left\{ \begin{array}{l} \mathcal{N}(\mu(x), \sigma(x)), \text{ if } z_j \in \mathbb{R}^{n_j} \\ w\mathcal{N}(\mu(x), \sigma(x)), \text{ if } z_j \in \mathbb{T}^{n_j} \end{array} \right.. \quad (2)$$

Here, the distribution-specific parameters $\mu(x)$ and $\sigma(x)$ are themselves input dependent, and given by a functional mapping which is learned using the reparametrization trick. We use a temporal (1D) convolutional neural network to parameterize $q$. To enforce circular latents, we let the encoder output vectors in $\mathbb{R}^2$, which we subsequently normalize to avoid discontinuities when estimating angles (Zhou et al., 2019). We remark that although a convolutional filter was used, temporal smoothness was not explicitly included in the latent dynamics of $z$ in the model used for producing the results in Sec. 4. It is, however, a straightforward inclusion to incorporate (see Appx. D.1).

For the decoder, in clear contrast to standard VAEs, we pick a simple parametric function $f$ to parameterize the reconstruction term $p(x|z)$. As argued above, this is a crucial choice in the interest of defining simple biologically meaningful link functions (i.e., tuning curves) between (potentially complex) latent variables and neural activity. Specifically, we have

$$p(x_i|z) \sim \mathcal{P}(f_i(z)), \qquad f_i(z) = \sum_j w_{ij} g_j(z_j, \theta_i), \quad (3)$$

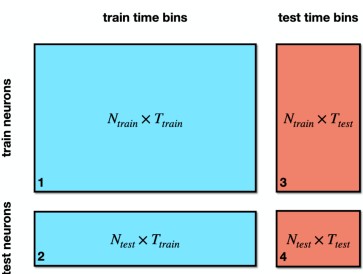

Figure 3: **Schematic outline of model evaluation**. Evaluation pipeline as suggested in Pei et al. (2021). During training, the models are all trained on spikes from 'train' neurons (**1**), learning their tuning curves, and the latent variable representation which is used to infer tuning curves on 'test' neurons at train time (**2**). For testing, we fix all tuning curve shapes and infer the latents from test data (**3**), before performing spike prediction on held-out data from held-out neurons (**4**). Thus, our encoder only receives train neurons (**1**, **3**) to infer latents.

where $\mathcal{P}(f_i(z))$ is a Poisson distribution with rate $f_i(z)$, and the weights for each neuron $i$ the output of a softmax function, such that $w_{ij} > 0$ and $\sum_j w_{ij} = 1$. That is, $w_{ij}$ models the participation of neuron $i$ in ensemble $j$. The function $g_j(z_j, \theta_i) \in \mathbb{R}$ characterizes the (shared) tuning curves of neurons in the latent space of $z_j$, and takes as input the corresponding latent $z_j$ and a (set of) neuron specific parameter(s) $\theta_i$, to produce the response of the neuron based on each latent space.

The function $g_j(z_j, \theta_i)$ can be specified in a multitude of ways, while still embodying the concept of feature sharing. Here we present two possible options: the first simply assumes a single heat kernel shape (i.e., Gaussian-like bump) with shared tuning width $\sigma$ for all neurons. Since neural activity is non-negative, we model the $\log$ of the instantaneous firing rates as

$$\log g_j(z_j, \mu_i) = -\frac{d_j(z_j, \mu_i)^2}{\sigma^2}, \tag{4}$$

where $\mu_i$ denotes (as above) the center of the $i$-th neuron's tuning curve and $d_j(\cdot)$ a distance function in the space of $z_j$ (i.e., Euclidean distance in $\mathbb{R}^n$ and geodesic distance in $\mathbb{T}^n$). The second, and more flexible, feature space consists of a sum of $M$ weighted ($\beta_m$) Gaussian basis functions (i.e., a spline)

$$\log g_j(z_j, \mu_i) = \sum_m \beta_m \exp\left[-\frac{d_j(z_j, \mu_m - \mu_i)^2}{\sigma_m^2}\right], \tag{5}$$

with center $\mu_m$ and width $\sigma_m$ of the $m$-th basis function. We consider three variations of faeLVM: sharing of features by utilizing Eq. 4 to model $g_j$ with a shared (bump) heat kernel (faeLVM-b), sharing of features through Eq. 5 using a Gaussian basis (faeLVM-s), as well as a model allowing each neuron to learn its own set of the basis weights in Eq. 5, i.e. no shared features (faeLVM-n).

## 4 Experiments

### 4.1 Simulations: Feature Sharing

Motivated by the evaluation pipeline in Pei et al. (2021), we split our data into four parts (Fig. 3), exploring the benefits of feature sharing in the following way: We generate $z \in \mathbb{S}^1$, a circular 1D latent variable and simulate spikes according to a Poisson distribution with heat kernel tuning curves (see also Appx. F for a retinal ganglion simulation with calcium dynamics). We fix $T_{\text{test}}$ and $N_{\text{test}}$ (1000 and 30), and investigate model performance based on fixed $T_{\text{train}}$ and varying $N_{\text{train}}$, and vice versa. For each condition, we repeat the experiment over 20 seeds (while data is different for each repeat, all models are trained on the same seed, ensuring valid comparisons), reporting mean negative log-likelihood (NLLH) and geodesic error (GE) on test data (lower numbers are better, in both cases). We also compare our models with mGPLVM (Jensen et al., 2020), an extension of GPLVM (Wu et al., 2018) designed for inference on non-Euclidean spaces, as well as a conventional VAE using a standard multi-layer perceptron decoder (MLP; with essentially a reverse architecture of the encoder). As mGPLVM assumes a Gaussian noise model, the Poisson NLLH is unsuitable as a measure, hence it is only included when evaluating latent recoveries.

Fig. 4A-B clearly show the regime where feature sharing excels: datasets with a satisfactory number of neurons, but with short recording times. This coincides well with the current state-of-affairs for neural recording techniques, where experiments are usually limited by the possible experimental length, not by the amount of recordable neurons (Stevenson & Kording, 2011). Moreover, even with further technical innovations, we may eventually be able to record from all neurons, but we will likely never be able to record during all possible natural inputs or behaviors.

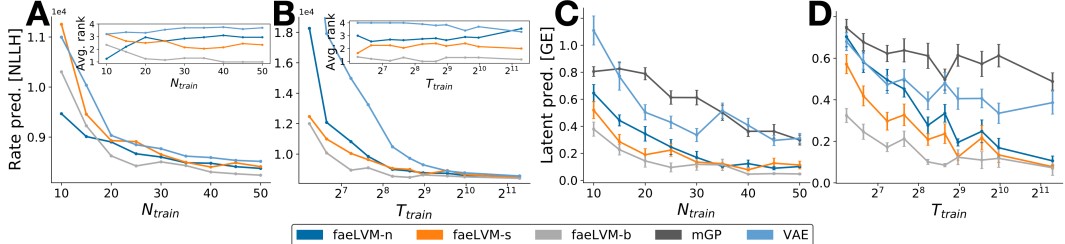

Figure 4: **Latent and Rate Prediction**. The models faeLVM-n,s,b are described in the main text, mGP is the model from Jensen et al. (2020), VAE a traditional approach using an MLP decoder. **A**, Mean NLLH of predicted test neuron rate, as a function of amount of training neurons. As the performance is averaged over 20 seeds of varying data, error bars would also reflect the variability in NLLH across data, resulting in inappropriate visualization. Hence, we report the mean rank of 20 seeds for each model at each condition, in an effort to more appropriately capture the variability of the models. **B**, Same as A, now as a function of training time points. **C**, Mean GE between true latent and inferred latent, on test data, with corresponding SEM error bars, as a function of number of training neurons. **D**, Same as C, now as a function of training time points.

We also remark that while faeLVM-n seems to performs better on rate prediction in the low neuron regime (Fig. 4A), the performance of LVMs ultimately hinges on how well the models can recover the correct latent variable. In the setting of latent recovery (Fig. 4C-D), we see that feature sharing is evidently beneficial both in low neuron- and low recording-time regimes. faeLVM-b outperforms the other models, perhaps somewhat expected given the shape of the synthetic tuning curves, while all faeLVMs outperform both mGPLVM and VAE. We also observe that while mGPLVM, which uses tuning curve decoders, is more interpretable, it is worse than neural network decoders. However, our approach with feature sharing closes that gap and even improves upon the existing VAE method, thus highlighting the importance of sharing features; more accurate inference, while also being interpretable. Note that although the faeLVMs leverage the benefit of a convolutional layer (which mGPLVM does not, being a non-convolutional model), we emphasize that the faeLVMs still outperforms mGPLVM, even after reducing the convolutional filter size down to 1, which forces the model to infer the latents independently at each time point (see Appx. E).

## 4.2 SIMULATIONS: ENSEMBLE DETECTION

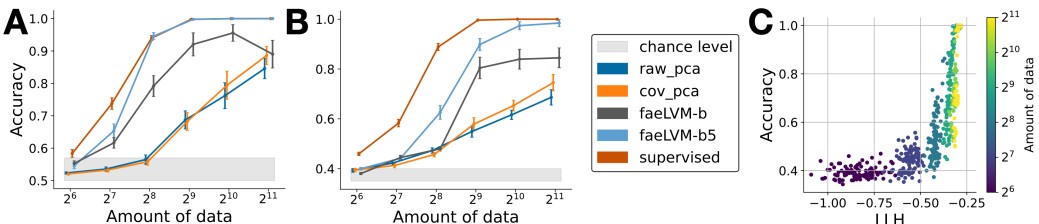

Figure 5: **Ensemble Detection**. Ensemble detection accuracy, with corresponding SEM error bars, on two (**A**) and three (**B**) equally sized ensembles of neurons tuned to separate $\mathbb{S}^1$ latent variables. The models are: k-means on PCA reduced data ('raw-pca'), k-means on PCA reduced covariance matrix of the data ('cov-pca'), supervised clustering based on the mutual information scores between the cell responses and the latent variables, faeLVM with shared heat kernel tuning curves ('faeLVM-b'), and the same model optimized over 5 seeds, selecting the one that produced the highest LLH ('faeLVM-b5'). Chance levels are computed from $100,000$ random labels with optimal permutations. **C**, Relation between likelihood and accuracy for different data amounts.

Next, we shift our focus to the challenge of ensemble detection, comparing the performance of faeLVM-b against other common methods. Specifically, we contrast against clustering methods (see Appx. C for more details) that either i) perform k-means clustering on dimensionally reduced neural activity over time (loosely inspired by Lopes-dos Santos et al., 2011; Baden et al., 2016; Hamm et al., 2021), or ii) perform clustering on the neural covariance matrix (loosely inspired by Carrillo-Reid et al., 2015). As an additional upper bound to the achievable performance, we include a *supervised*

clustering method that receives both the mutual information score between each neuron's activity, and the true latent variables, as input. We run our model over five different random seeds and pick the run that yields the highest LLH (faeLVM-b5 in Fig. 5), a fair approach as the likelihood is an unsupervised metric (k-means is run over 100 seeds, picking the one with the lowest inertia).

The setting we chose for these experiments was the separation of ensembles of neurons that live on a ring, i.e. $\mathbb{S}^1$. The comparison methods failed to separate the ensembles when scaling up to neural activity on a torus, in contrast to faeLVM, which further highlights the novelty of an unsupervised method that can accomplish this task (see Sec. 4.4 for tori, and also Appx. I for an additional ensemble separation ablation). We tested the accuracy on separating two and three distinct (no shared latents across ensembles) neural ensembles of equal size on rings, in addition to investigating accuracy for all models as a function of available data points (recording length).

Fig. 5A-B shows that our model outperforms both comparison methods and quickly approaches the performance of the supervised upper bound, for both two and three rings. This is even more pronounced for the model selected based on LLH — a viable option when the analysis result warrants additional computing time. Fig. 5C shows that the relationship between ensemble detection accuracy and LLH is monotonically increasing. Thus, the unsupervised likelihood score is indeed a good proxy to arbitrate between different optimization runs.

## 4.3 REAL DATASETS: HEAD DIRECTION DATA

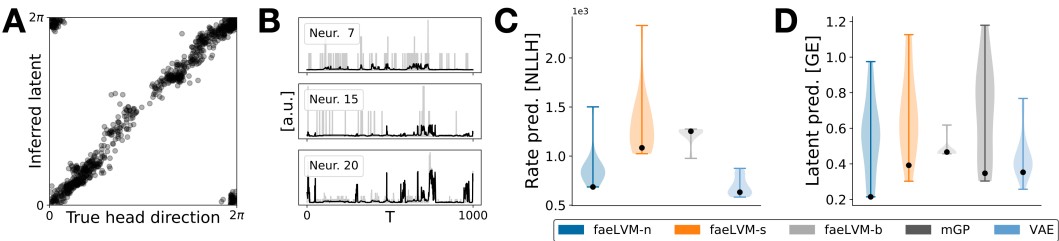

Figure 6: **Application to head direction data**. **A**, Variational mean of faeLVM-n plotted against recorded mouse head direction (for additional visualizations, see Appx. H). We trained 20 models, selecting the one with highest training likelihood on training neurons, before performing inference at test-time. **B**, True test neuron spikes (grey), and predicted test neuron rates (black) from the model used in A, per bin, for three randomly selected neurons. **C**, NLLH of test neuron rate, over 20 seeds, as a violin plot. Performance of the best model, based on train LLH, is indicated by the black dot. **D**, Same as C, but comparing GE between recorded head direction and inferred latent, on test data.

Transitioning to real datasets, we first apply our models to data from Peyrache et al. (2015b); more specifically the awake trials from the dataset labeled Mouse 28, session $140313$, recorded from the anterodorsal thalamic nucleus (ADN). Recordings yield approx. $35$ min. of data, which we bin in $100$ ms bins, before separating it with a train-test split of $95\% - 5\%$. 3 neurons are allocated as test neurons (out of 26 total) and we run 20 seeds of each model, selecting the best performing model for visualization (faeLVM-n). The inferred latent (Fig. 6A) is observed to match the true head direction to a high degree, and the model is able to predict rates on test neurons (Fig. 6B), even though the average and peak activity over test bins are quite low ($\sim 1$ and $5$ Hz respectively) for the first two.

As we can see from Fig. 6C, the unconstrained model (faeLVM-n) outperforms the other two. This is not an unsurprising result, given the model's higher complexity and the sufficiently large amount of training data. As for the latent prediction, mGPLVM and the traditional VAE (Fig. 6D) are close in comparison to the performance of faeLVM-s, although less accurate than faeLVM-n. We also note that the VAE does perform better on the rate prediction, which, conceivably, might be explained by the fact that it is not constrained by a particular tuning curve model and thus might learn interactions that incorporate non-head direction variables in an effort to improve its rate prediction. Training multiple models and making a selection based on train LLH generally correlates well with solid results on the test set, as also shown in Fig 5C. Note that faeLVM-b achieves much more consistent optimization results over different random initializations, suggesting a typical bias-variance tradeoff.

Overall, we recognize that while there are differences in performances, all models recover the correct circular manifold. The faeLVMs however, are noticeably faster ($\sim 5$ min. run time for one seed, on

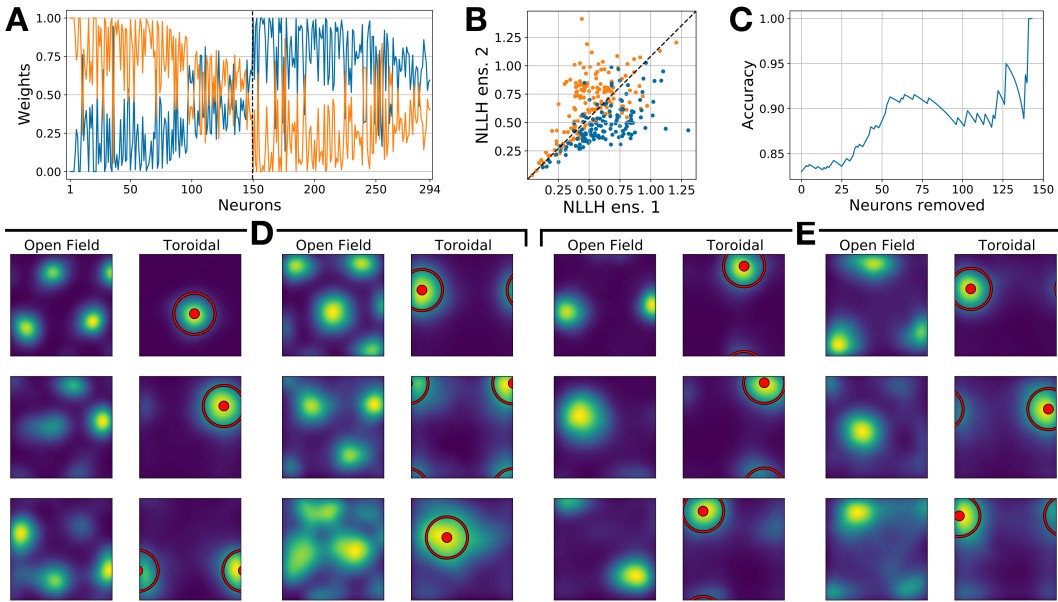

Figure 7: **Application to grid cell data**. **A**, Learned ensemble weights. Neurons are ordered by decreasing spatial information, dashed line separates first and second reported module (Gardner et al., 2022). **B**, Greedy test for each neuron, checking which latent provides higher likelihood (i.e., one-hot ensemble weights), colored by reported module. **C**, Classification accuracy when successively removing neurons with low spatial selectivity. The same number of neurons were removed from each ensemble, starting with least spatially informative ones. Accuracy is measured against supervised ensemble detection from Gardner et al. (2022). **D**, Rate map of every $25^{th}$ neuron, first grid cell module (smoothed for visualization purposes). Maps shown as function of rat's position in arena, as well as inferred coordinates on latent torus. Red circles indicate inferred receptive field mean and width (SD) for the heat kernel (i.e., Gaussian-like bump). **E**, Same as D for second module. For a full set of rate maps, as well as latent dynamics, see Appx. G & H.

a CPU), compared to mGPLVM ($\sim$ 50 min. run time for one seed on a GPU). While not critical in the case of this particular dataset, it is paramount when inferring latents of higher dimensionality. We also note that while the circular manifold was in this case known, it is possible to do model selection over different latent spaces (toroidal, circular, planar, etc.) as was demonstrated in Jensen et al. (2020). Thus, it is not strictly necessary to know the latent topology when applying our method, rather it allows for the restriction of possible models to a limited hypothesis class.

## 4.4 REAL DATASETS: GRID CELL DATA

Finally, we apply the complete model pipeline to data recorded from the medial entorhinal cortex (MEC) of a freely moving rat (Gardner et al., 2022). Neurons in the MEC are selective to a number of features like space (Fyhn et al., 2004), head direction (Sargolini et al., 2006), speed (Kropff et al., 2015), as well as conjunctive representations of the three (Sargolini et al., 2006; Hardcastle et al., 2017), though most demonstrate a strong preference for single features (Kropff et al., 2015; Tukker et al., 2022). We consider data from the rat R, day 1-session, using recordings from neurons labeled module 2 and 3, considering 149 and 145 non-conjunctive neurons respectively. Spike data was binned in 100 ms bins, after removing periods in time when the rat was considered stationary (speed $< 2.5 \, \text{cm s}^{-1}$). Recordings from both the open field and maze experiments yields approx. 170 min. of data. We train 20 models from different seeds, selecting the one with highest in-sample likelihood.

Compared with the modules discovered in the dataset (Gardner et al., 2022), our model's results coincides with an accuracy of $0.83$. The inferred weights (Fig. 7A) show a clear separation between the two modules, becoming less distinct as the neurons' spatial information decreases. This is also reflected in Fig 7B, where neurons further away from the diagonal tend to be more accurately classified. However, we note that the original separation of neurons into ensembles is not an intrinsic truth; a number of the less informative neurons exhibit structurally ambiguous rate maps (see Appx. G), a possible indication that they may belong to a different ensemble altogether. Excluding these neurons

while evaluating the model (Fig. 7C), we observe a significant increase in performance, reaching 0.90 accuracy after removing approx. 50 of the least informative neurons from each ensemble.

Fig. 7D-E show a selection of rate maps for neurons from both ensembles. We see that the model infers toroidal decodings that keep the structural integrity of the receptive fields, akin to the findings in Gardner et al. (2022). The inferred centers and widths also correspond well with the rate maps. We note that, while promising, the dataset included here contains just 294 of the 2460 neurons originally recorded and that the full dataset (currently not available) would require additional steps to determine the number of ensembles, the corresponding shapes and allow for some degree of conjunctivity.

## 5 CONCLUSION

In this paper we provide a proof-of-concept that the paradigm of share and divide is a successful application of the old idea of *carving nature at the joints*,[1] in the sense that similar observations are grouped while differences are highlighted. This principle, instantiated in a group equivariant computation core and a cell-type specific readout, has been successfully deployed in visual neuroscience; we show that it also provides an excellent inductive bias (or prior) in the case of higher cognitive variables such as (but not necessarily limited to) head direction or position in space. It is advantageous from a statistical perspective to leverage the similarity in tuning functions across neurons, as well as from a biological point of view, as finding protoypical tuning curves can help us unravel the common tuning properties across individual neurons.

The idea of *separating ensembles* is at the core of the scientific endeavor where the task is to group our observations of nature into distinct categories. In the case of grid cell modules, it is absolutely crucial, as the topological properties that define this class of neurons are completely hidden when the separate ensembles of different spatial resolution are not properly distinguished. One might wonder whether other brain areas such as primary visual cortex, for which a power law in the population activation space has been stipulated (Stringer et al., 2019), may not resolve into simpler underlying topological ensembles of neurons, such as the special orthogonal group of three-dimensional rotations $SO(3)$ (vision), 2-spheres $\mathbb{S}^2$ (Singh et al., 2008), or the Klein bottle shape of oriented and phase shifted wavelets (Carlsson et al., 2008) (relevant to multiple sensory modalities).

Furthermore, the assumption of *feature sharing* relies on an equivariant neural code, such as the approximate translation equivariance in the visual system, which certainly is an abstraction from the actual inter-neural variability that exists in the real world. In some cases, it might be that neurons are better described along some continuous space of variations, rather than by a fixed set of discrete prototypical tuning curves (Ustyuzhaninov et al., 2022). On the other hand, if we can identify the parameters (e.g., tuning width) that describe the continuous variations across neurons and separate them from the stable features of tuning curves (e.g., Gaussian-like heat kernel shape), we can use that to build a more flexible parametric (but still shared) feature basis. Examples of such approaches include rotation (Ustyuzhaninov et al., 2019), kinetic (Zhao et al., 2020) or more general nonlinear modifications (Shah et al., 2022) to shared feature spaces. Further challenges to the feature sharing idea could arise from recent findings of representational drift (Rule et al., 2019) and location- (Qiu et al., 2020) or context dependent tuning (Kanter et al., 2017), although future directions might consider addressing these in the model. Other limitations are the fact that the grid cell torus is hexagonal, while we use an orthogonal basis; further analyses might reveal if the encoder compensates for that, e.g., by adjusting the distribution of speeds on the torus per direction.

Finally, although the choice of variational posteriors might at first seem restrictive, we would argue that the corresponding topologies (circles, tori, $\mathbb{R}^n$) are natural choices for latent spaces and likely to keep appearing in the brain, as is suggested in recent work (Kriegeskorte & Wei, 2021). Other authors conjectured that, e.g., prefrontal cortex deploys grid cell-like codes to explore physical (Doeller et al., 2010; Jacobs et al., 2013) and conceptual spaces (Constantinescu et al., 2016; Bao et al., 2019), as well as rules in a reinforcement learning task (Baram et al., 2021). The challenge of constructing or observing the relevant space has proven a hurdle for previous supervised methods, being unable to discover cognitive maps without *a priori* knowledge of the represented covariates. Our model is particularly suited in this setting, offering a targeted method to identify neural ensembles and summarizing as well as leveraging their shared tuning properties.

---

[1]Plato (428-348 BC), *Phaedrus* 265e; also, Zhuangzi (369-286 BC), *Nourishing the Lord of Life*.

ETHICS STATEMENT

More generally, reducing high dimensional nonlinear dynamical systems to a set of latent variables can have a broad array of possible applications, for instance, in climate sciences, crop forecasting or flood prediction. Restraining the decoders to interpretable mechanisms and groups of observations could, also in those cases, help us further our understanding of the underlying processes or relations between low dimensional summaries and data observations. As for any complex and versatile modern machine learning method, there also exists a danger of translating the insights from this work to detrimental purposes and intents. Applications to sensitive data with protected attributes, such as gender or ethnicity, should pay special attention to the en- and decoding of those attributes in the learned latent spaces. Therefore, while we do not see any obvious misuse, nor want to explicitly name any possible malicious purposes, we still strongly discourage any nefarious applications of the ideas developed in this work. Lastly, we tried to minimize the environmental impact of our research by performing the hyper-parameter search on the GPU cluster of a carbon-neutral organization.

REPRODUCIBILITY STATEMENT

Code and implementation are included as supplementary material, and is also made available at: https://github.com/david-klindt/NeuralLVM. Datasets are public, and details regarding model parameters, data generation, etc. are specified under the relevant sections (e.g., Sec. 4), as well as more thoroughly in the Appendix (e.g., Appx. A, B, C, J & K).

ACKNOWLEDGMENTS

We thank Erik Hermansen for valuable discussions and feedback related to the grid cell data, as well as Ta-Chu Kao for fruitful discussions. This work was supported by a Norwegian Research Council Large-scale Interdisciplinary Researcher Project Grant iMOD (NFR grant no. 325114).

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

APPENDIX

## A  MODEL DETAILS

All models are trained with the Adam optimizer (Kingma & Ba, 2014), a learning rate of 0.001, temporal chunks of size 128 (64 if data length is smaller than 128) and a batch size of 1. Moreover, training is concluded after the training objective has not improved for 5 steps (10 for grid cell data). All models are implemented in Pytorch (Paszke et al., 2019). The architecture of the encoder is as follows: a 1D grouped convolution with kernel size 9; followed by two layers of 1D convolutions with kernel size 1 (i.e., effectively just MLPs at each time point) and 64 filters (128 for grid cell data), followed by a linear projection onto the dimensionality of the mean and variance parameters required by the variational posterior. The decoder follows the feature sharing tuning curve structure as presented in the paper. All experiments where executed with specified random seeds for data generation, model initialisation and training batch selection to maintain complete reproducibilty.

### A.1  A FAST APPROXIMATION TO THE OBJECTIVE FUNCTION

Next, we present our simplification of the theoretical setup from (Davidson et al., 2018), concerning the use of von Mises distributions when inferring latent variables on circular manifolds. First, we note that as the variance $\sigma^2$ of a Gaussian and the inverse concentration $1/\kappa$ of a von Mises approach 0, they both become delta functions. Moreover, they each define the maximum entropy distribution for a fixed variance (concentration) in their respective domains. Hence, the von Mises is also referred to as the circular normal distribution.

We therefore explored the model behavior when replacing the KL term in Eq. (11) with a simpler expression, akin to the standard VAE KL, which simply pushes the variance of the variational encoder towards 1 (i.e., standard normal), but without constraints on the mean (uniform over the latent space, as the full von Mises prior). Reparameterisation and sampling was then performed by drawing from that Gaussian and adding the scaled perturbation (same as in standard VAEs) to the angle that the variational posterior returned as mean estimate.

In general, performing variational inference in non-Euclidean spaces is challenging (Falorsi et al., 2019), which has warranted the use of specialized inference procedures in previous latent variable models on such manifolds (Davidson et al., 2018; Falorsi et al., 2019; Jensen et al., 2020; 2022). In this work, we follow the 'ReLie' approach outlined by Falorsi et al. (2019) for general Lie groups, which involves defining a parameterizable (in our case Gaussian) distribution on the tangent space of the group and projecting it onto the manifold using the so-called 'exponential map' of the group. This approach is particularly simple for the toroidal manifolds we consider, where the projection step can be achieved by a simple modulo $2\pi$ operation (Jensen et al., 2020). This leaves the challenge of computing the KL term in Eq. 11, which is given by

$$\mathrm{KL}[q(z)|p(z)] = E_{q(z)}\left[\log q(z) - \log p(z)\right]. \tag{6}$$

As a prior $p(z)$, we use a unit Gaussian projected onto the circle, with a mean matched to the posterior mean. Since the KL divergence in Eq. 6 is invariant to a rotation around the circle (Falorsi et al., 2019), we can compute it simply as the KL divergence between two wrapped Gaussian distributions centered at the origin with variances $\sigma_q^2$ and $\sigma_p^2 = 1$ respectively. In general, the projected densities needed in Eq. 6 are given by

$$q_\theta(z) = \sum_{x \in \mathbb{R}\,:\,\exp_G(x)=z} r_\theta(x)|J(x)|^{-1}, \tag{7}$$

as derived in Falorsi et al. (2019). Here, $\exp_G$ is the exponential map, $J(x)$ is the Jacobian at $x$ in the tangent space, and $r_\theta$ is the parameterized reference distribution on the tangent space.

For the circle, the Jacobian is simply given by the identity $J(x) = 1 \forall_x$. With a zero-centered Gaussian reference distribution, we can therefore rewrite Eq. 7 as

$$q_\theta(z) = \sum_{k \in \mathbb{Z}} \mathcal{N}(z + 2k\pi; \mu = 0, \sigma_q^2), \tag{8}$$

and similarly for the prior $p(z)$ (Jensen et al., 2020). To approximate this infinite sum, it is necessary to truncate it after a finite number of elements (Falorsi et al., 2019; Jensen et al., 2020; 2022). In the case where $\sigma_q \ll 1$, it is sufficient to use a single term since there is negligible probability mass outside the area of injectivity ($[\pi, \pi]$). This is in fact the regime we are operating in, since less than $0.2\%$ of the prior probability mass falls outside the interval $[-\pi, \pi]$, and the posterior variance will in general be smaller than the prior variance.

Additionally, while the integral in Eq. 6 covers the domain $[-\pi, \pi]$, we can approximate it with an integral from $-\infty$ to $\infty$ since the posterior probability mass outside the area of injectivity is negligible, $\int_{x=\pi}^{\infty} \mathcal{N}(x; 0, \sigma_q^2) dx \approx 0$. These approximations together imply that we can simply approximate the KL term in Eq. 6 as the KL between our prior and posterior Gaussians *directly in the tangent space*, and we verified numerically that this approximation is excellent for $\sigma_q^2 \leq 1$. We thus obtain a simple analytical expression for the KL:

$$\mathrm{KL}[q(z)|p(z)] \approx \mathrm{KL}\left[\mathcal{N}(z; 0, \sigma_q^2)|\mathcal{N}(z; 0, 1)\right] \tag{9}$$

$$= \frac{1}{2}\left(\sigma_q^2 - 1 - \log \sigma_q^2\right). \tag{10}$$

Note that this is only possible because our prior is *localized*, in contrast to previous work which often uses a more diffuse or even uniform prior distribution (Falorsi et al., 2019; Jensen et al., 2020), and because we consider circular latent spaces where the Jacobian is 1. Since our prior and posterior both factorize across dimensions, these considerations also generalize to higher dimensional spaces, where the multivariate KL divergence is simply the sum across dimensions of the corresponding univariate KL divergences.

Finally, to verify that these approximations do not affect our results or conclusions, we repeated several key analyses with the variational inference framework described in Davidson et al. (2018). The simplified setup trained about $5\times$ faster, while behaving nearly identically on all test cases that we evaluated, hence, all experiments in the paper are performed with the simplified objective. We note that this simplified procedure may be of interest to those who might need to iterate over models without access to a large compute cluster.

## B  DATA GENERATION FOR SIMULATIONS

When conducting the experiments that showcase the scaling of various models with respect to the amount of data (Sec. 4.1), we generate the spikes according to Gaussian-like tuning curves. More specifically, we assume a common shape for each curve (location assigned randomly on $\mathbb{S}^1$), with a tuning width of 1.2 radians. Peak rates are set to 0.5 spikes per bin, the background rate to 0.005, with a signal-to-noise ratio of 1 for the Poisson noise. The latent variable is simulated according to a Gaussian process with kernel standard deviation equal to 5.0, and with kernel scale 50.0. It is then projected onto $\mathbb{S}^1$ via the modulo operation. The ensemble detection experiments in Sec. 4.2 use the same data generation, only with multiple (2 or 3) ensembles, with distinct latent varibles for each ensemble.

## C  ENSEMBLE DETECTION COMPARISONS

For *raw-pca*, we performed k-means clustering on dimensionally reduced neural activity over time (loosely inspired by Lopes-dos Santos et al., 2011; Baden et al., 2016; Hamm et al., 2021). Specifically, we took the projection of the spike matrix onto its first 8 (cross-validated) principal components. We then computed k-means clustering on this reduced matrix with the standard scikit-learn implementation, using a fixed number of 2 (3, depending on the setting) clusters, 100 initializations and 1000 maximum iterations.

For *cov-pca*, we performed clustering on the neural covariance matrix (loosely inspired by Carrillo-Reid et al., 2015). Specifically, we computed the covariance across neurons from the spike matrix. Again, we computed the first 8 (cross-validated) principal components of the absolute value of this covariance matrix. The absolute value was taken to preserve any negative or positive dependencies across neurons, as one would presumably observe within an ensemble — removing the modulus operator drastically reduced ensemble detection accuracy. From the projection onto the principal components we proceeded again as above. We also performed a clustering of the inverse of the

covariance matrix, as a first order approximation to an Ising model (Schneidman et al., 2006), but this approach did not yield results above chance level, hence it was not included in the figures.

Finally, for the *supervised* method we performed clustering on the mutual information score (see scikit-learn for implementation) between each neuron's activity, and the true latent variables, as input. Specifically, the features $y$ that were used as input to k-means (same settings as above) were $y_{ij} \sim I[x_i; z_j]$ for every neuron $i$ and every *ground truth* latent dimension $j$. This presents an upper bound to the achievable performance, since it assumes *supervised* knowledge of the true latent variable values.

## D    THE EVIDENCE LOWER BOUND (ELBO)

Here we show the derivation of the ELBO, which is maximized as part of the objective function during the model training procedure

$$
\begin{aligned}
\log p(x) &= \log \int p(x|z)p(z)dz = \log \int p(x|z)p(z)\frac{q(z|x)}{q(z|x)}dz \\
&= \log E_{z\sim q(z|x)}\left[p(x|z)\frac{p(z)}{q(z|x)}\right] \overset{\text{Jensen's inequality}}{\geq} E_{z\sim q(z|x)}\left[\log p(x|z)\frac{p(z)}{q(z|x)}\right] \quad (11) \\
&= E_{z\sim q(z|x)}\underbrace{[\log p(x|z)]}_{=:\mathcal{L}_{\text{rec}}(x,z)} - \underbrace{E_{z\sim q(z|x)}[\log q(z|x) - \log p(z)]}_{= \text{KL}(q(z|x)|p(z))} =: \text{ELBO}(q).
\end{aligned}
$$

### D.1    TEMPORAL TRANSITION PRIORS

All experiments in the article are conducted using the objective function from Eq. 11. Extending this with additional terms however, e.g. by including a Laplacian temporal transition prior (Klindt et al., 2021) or a Gaussian transition prior (i.e. an $L_1$ or $L_2$ penalty across latent time steps, respectively) is relatively simple, and both the $L_1$ and $L_2$ penalties are included as optional regularization terms in our model. While exhaustive experiments have not been performed regarding these priors, results seem to indicate that the inclusion of these priors do little to improve the model performance in the settings we have studied in the article, which is in agreement with results in Appendix Section E and K regarding the convolutional kernel size.

## E    ABLATION STUDY — CONVOLUTIONAL FILTER

As mentioned in Sec. 4.1, we conduct a similar model performance comparison between the faeLVMs and mGPLVM, while reducing the kernel size of the convolutional layer for the faeLVMs to 1 (effectively allowing the encoder network to let the variational posterior only depend on the instantaneous neural population activity). This is done to reduce the potential gap in information content available between a convolutional and a non-convolutional model. Note, however, that this applies only to the variational posterior. At inference time, both models infer the best latent at each point in time separately. Experimental settings are precisely the same as those in Sec. 4.1, other than the size of the convolutional kernel.

One of the main observations from Fig. 8C-D is that the faeLVMs still outperform mGPLVM, even without contributions from the convolutional kernel. In fact, faeLVM-s performs much better now, compared with the results from Fig. 4, achieving results that rivals those of faeLVM-b (Fig. 8A-B)

## F    MODEL EXTENSION — CALCIUM RESPONSES AND VISUAL DATA SIMULATION

In an effort to improve on the applicability of faeLVM, and showcase its relevance regarding applications to different brain regions, different latent topologies and different input data, we include a toy example on visual data, where we modeled a moving dot stimulus (Fig. 9A) and a population response of retinal ganglion cells with center surround receptive fields and Poisson spiking (Fig.

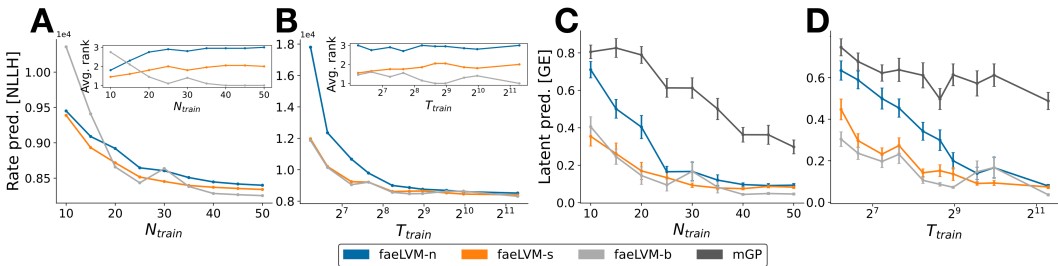

Figure 8: **Model performance comparison, with convolutional filter size** 1. The models faeLVM-n,s,b are described in the main text, mGP is the model from Jensen et al. (2020). **A**, Mean NLLH (negative log-likelihood, lower is better) of predicted test neuron rate, as a function of amount of training neurons. As the performance is averaged over 20 seeds of varying data, error bars would also reflect the variability in NLLH across data, resulting in inappropriate visualization. Hence, we report the mean rank of 20 seeds for each model at each condition, in an effort to more appropriately capture the variability of the models. **B**, Same as in A, now as a function of training time points. **C**, Mean GE (geodesic error, lower is better) between true latent and inferred latent, on test data, with corresponding SEM error bars, as a function of number of training neurons. **D**, Same as in C, now as a function of amount of training time points.

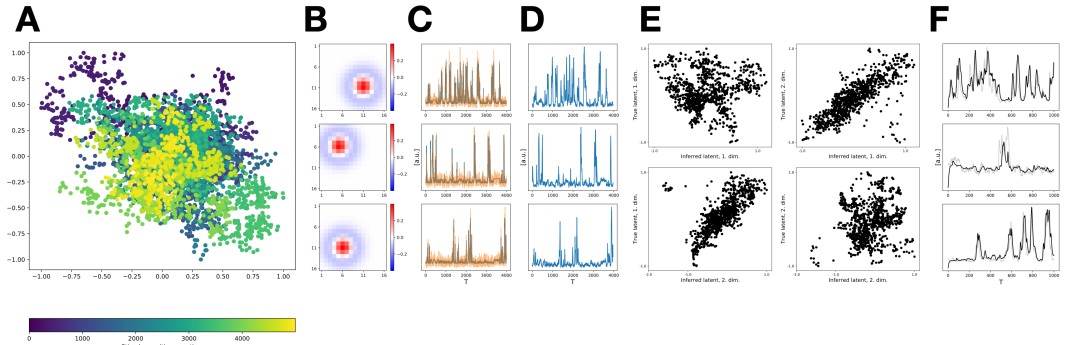

Figure 9: **Visual data, toy example**. **A**, Scatter plot of trajectory of simulated visual stimulus, colored according to its temporal evolution. **B**, Receptive fields for three training neurons. **C**, Same three training neurons, showing firing rate in shaded dark blue and the corresponding Poisson spikes in orange. **D**, Neural responses after passing the neural activity from C through a calcium kernel. **E**, Scatter plot of true latent trajectory against inferred latent trajectory (variational mean). **F** True Ca2+ neural responses (grey) plotted against predicted neural responses for three test neurons (black).

9B-C). Each neuron's spiking response was then convolved with a double exponential calcium fluorescence kernel (Fig. 9D). We further extended faeLVM by including a Gaussian likelihood as a modeling option, as well as an option to infer latent trajectories not restricted to circular and toroidal latent manifolds.

Results can be seen in Fig. 9E-F, where the model is able to infer the correct two-dimensional latent variable, as well as accurately predict calcium responses of test neurons on test data. Although this experiment is less comprehensive, being a toy example, it still gives a clear indication that our model is also applicable to both calcium traces and non-grid and head direction cells.

## G   ALL GRID CELL RATE MAPS

In this section, we showcase rate maps for all neurons in each of the two ensembles (149 and 145 neurons in the first and second grid cell module respectively). The data used is the same as discussed in Sec. 4.4, pre-processed in the same way, and we used the same model to produce latent decodings, receptive fields and ensembles.

Fig. 10 and Fig. 11 show the rate maps (using $50 \times 50$ spatial bins, smoothed for visualization purposes in the same manner as Gardner et al. (2022), with a Gaussian kernel of smoothing width

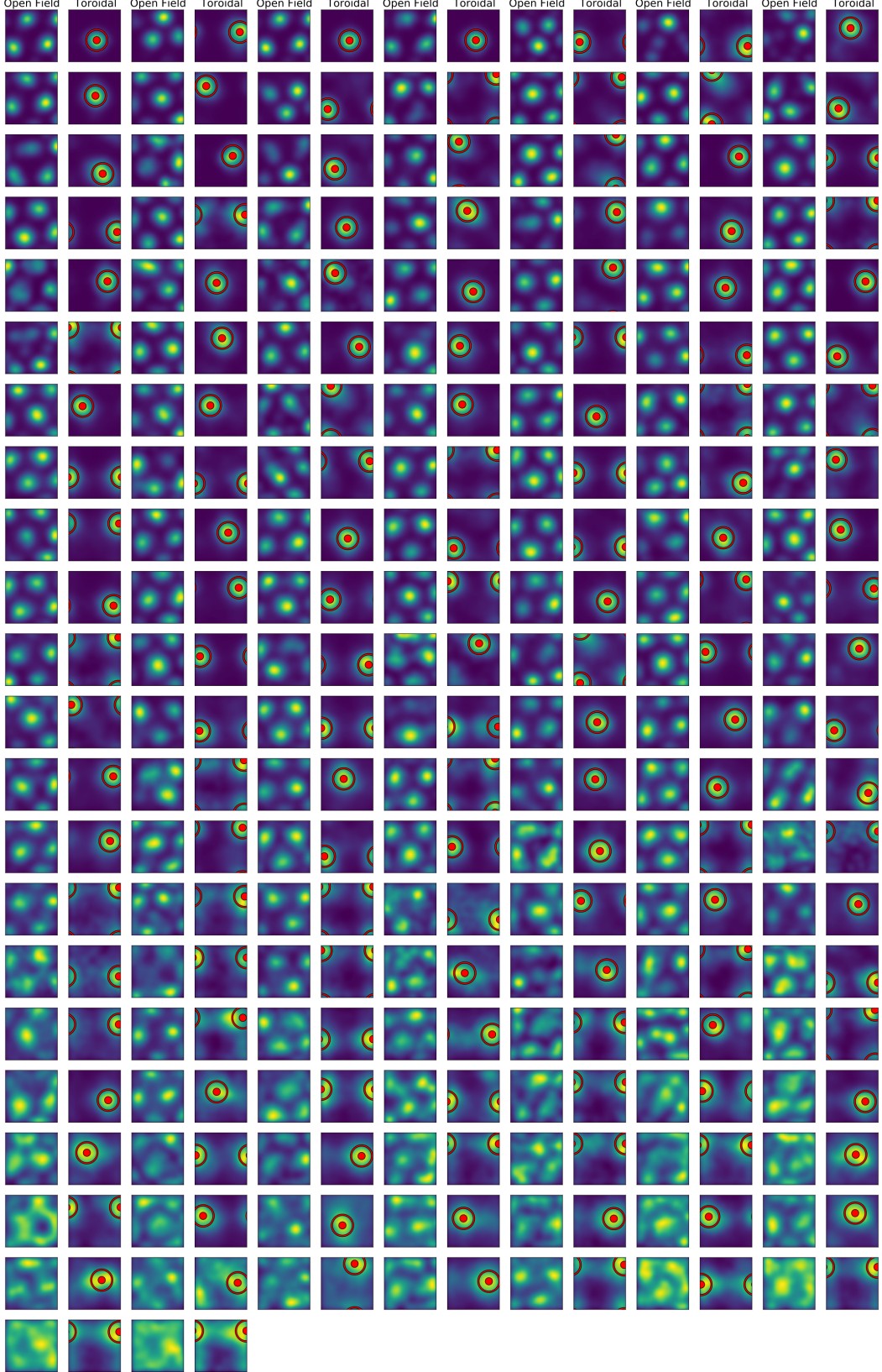

Figure 10: **All rate maps, first grid cell module**.

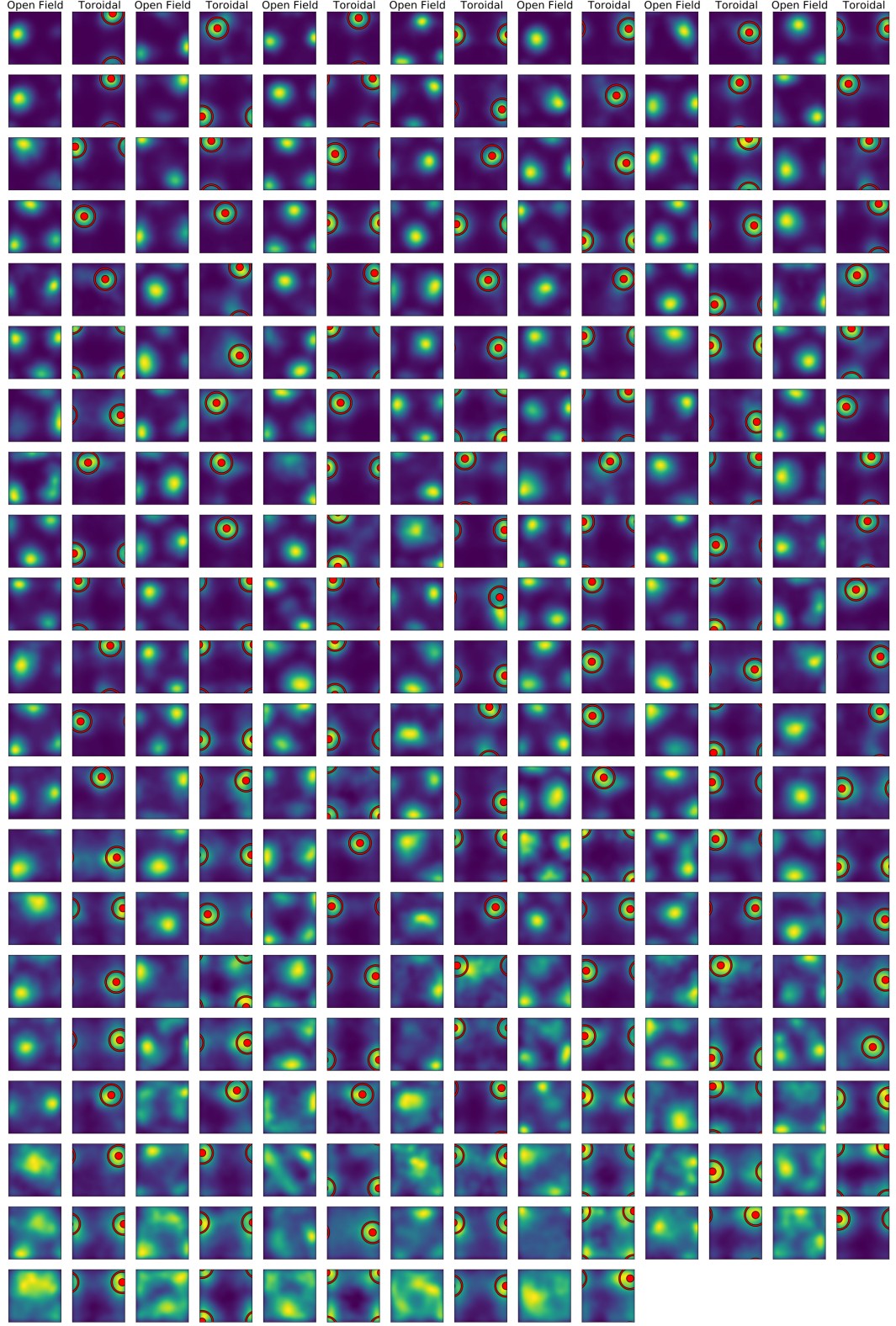

Figure 11: **All rate maps, second grid cell module**.

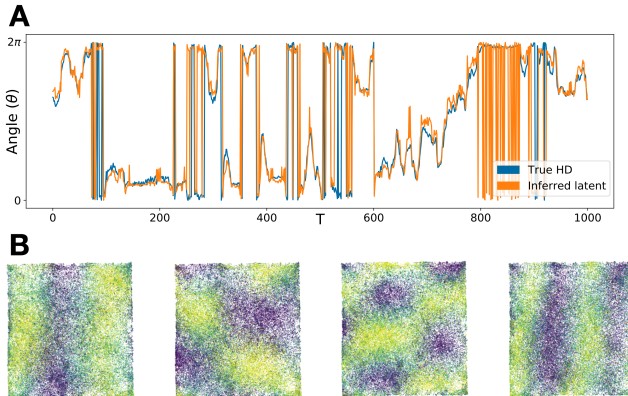

Figure 12: **Supplementary visualizations for analyses on real datasets**. **A**, Recorded mouse head direction ('True HD') and inferred latent trajectories on test data. Results from the experiment on head direction data, Sec. 4.3. **B**, Dynamics of the inferred toroidal coordinates plotted as a function of the $(x, y)$ spatial position of the animal (two tori, each with two circular coordinates, hence four figures). Results from the experiment on grid cell data, Sec. 4.4.

2.75), both as function of spatial position and inferred toroidal coordinates. For the rate maps in torodial coordinates, we also include the inferred receptive field center and width. Neural rate maps are shown in decreasing spatial information content (Skaggs et al., 1992; Gardner et al., 2022), going from left to right, top to bottom.

As mentioned in Sec. 4.4, we clearly see the more ambiguous rate maps at the bottom of these two figures (in particular, the three-four bottom-most rows). Here, the spatial resolution is not as distinct as for the more informative neurons, and the rate maps less clear, for which one might surmise that these neurons need not necessarily belong to the module originally assigned to.

## H    ADDITIONAL VISUALIZATIONS FOR HEAD DIRECTION AND GRID CELL DATA

To supplement the results shown in Sec. 4.3 and Sec. 4.4, we have included additional figures for both the head direction application and the grid cell application. More precisely, Fig. 12A shows the inferred variational mean of faeLVM-n plotted against the recorded mouse head direction (now as trajectories, in contrast to the scatterplot in Fig. 6A, while Fig. 12B shows inferred latent dynamics on grid cell data, as function of spatial position. As there is no way of recording the 'true' toroidal coordinates (akin to what is done in the head direction dataset), this figure is included in an effort to demonstrate the clear correspondence between each of the inferred toroidal axes and the spatial position, with the results being in agreement with what was found in Gardner et al. (2022).

## I    EXTENDED ENSEMBLE DETECTION SIMULATION

In this section we present an additional experiment related to the ensemble detection task. Namely, we ask the model to separate five distinct ring-like ensembles, $z_j \in \mathbb{S}^1$, to complement the experiments from Section 4.2, which only considered two and three ensembles.

From Fig. 13, we can see that faeLVM-b is able to separate the five ensembles quite clearly. The weights in Fig. 13A is approaching the desired one-hot encoding, and we see that the five inferred ensembles correspond very nicely with the true ensembles (Fig. 13C). The model is also able to accurately predict spike rates on held out neurons from each ensemble (13B). Thus, this simulation provides evidence that our model is also able to work on more challenging ensemble separation tasks, and is not only limited to lower ensemble numbers.

## J    HYBRID INFERENCE

### J.1    OVERVIEW

Here, we give an overview on how the hybrid inference is performed. In contrast to the traditional variational approach, where one uses a single sample from the variational posterior as prediction for the latent variable, we propose drawing a selection of samples, then performing gradient descent on said latents using the Adam optimizer. Specifically, if $z$ are the latents for the test set (as inferred by

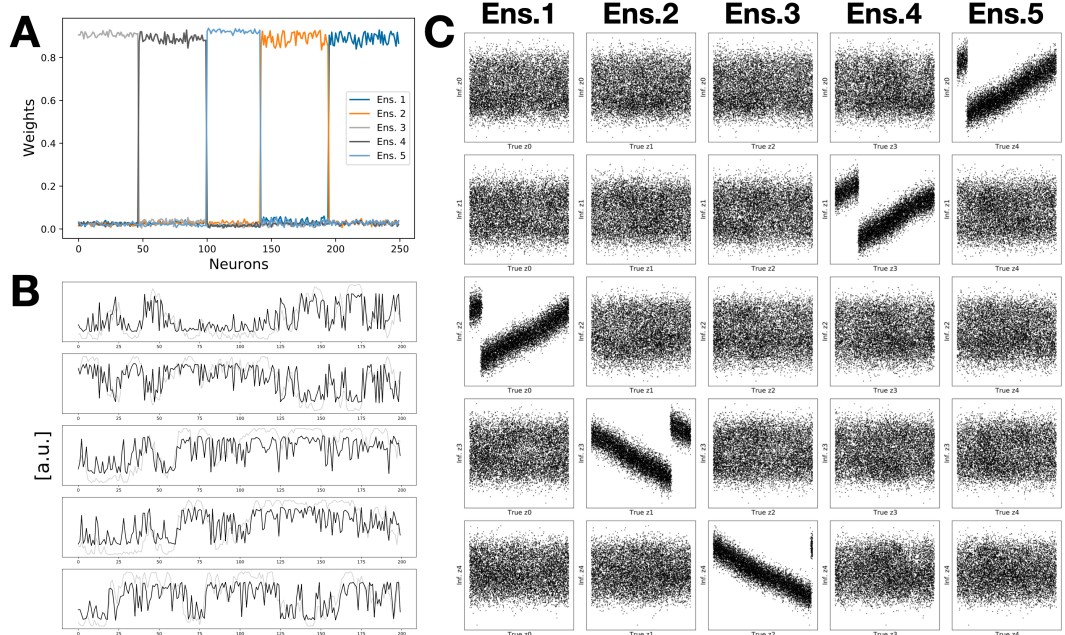

Figure 13: **Ensemble detection of 5 circles**. **A**, Learned ensemble weights for the neurons belonging to five unique circular ensembles. **B**, True spike rate (grey) vs predicted spike rate (black) for for five randomly selected test neurons, one from each ensemble. **C**, Scatter plot of true latent trajectory against inferred latent trajectory.

the encoder and variational posterior) and $x$ the responses of the training neurons on the test set, we maximize

$$z^* = \max_z \log p(x|z) \tag{12}$$

for 2000 steps with the Adam optimizer and a learning rate of $0.001$. We perform this in parallel on $M = 10$ distinct samples from the variational posterior and pick the one with the highest final likelihood (note that all samples usually converge to the same value). We emphasize that all model parameters are fixed during hybrid inference, and the optimization only affects the sampled latents.

Empirically, we observe that this not only improves the likelihood for the training neurons (i.e., the objective we are maximizing), but crucially also the likelihood of the test neurons. This means that we are not just overfitting but actually improving our point estimates of the latents in this inference procedure. We leave optimization of the full likelihood, including uncertainty estimates on the latents, for future research.

### J.2 HYBRID INFERENCE SIMULATIONS

For evaluate of the performance gain with hybrid inference at test-time, we consider six variations of the faeLVM; that being the three cases described in Sec. 3, each alternative including a case of using hybrid inference and one using the standard variational posterior approach (fae vs v-fae).

As in Sec. 4.1, we divide our data into four parts (Fig. 3), generate a 1D periodic latent variable and simulate spikes according to a Poisson distribution with Gaussian tuning curves. We fix $T_{\text{test}}$ and $N_{\text{test}}$ (1000 and 30), vary $T_{\text{train}}$ and $N_{\text{train}}$ over a selection of values, and for each condition, repeat the experiment over 20 seeds (synthetic data is different for each seed, but all of the models are trained on the same dataset, to ensure that comparisons are still valid), reporting mean negative log-likelihood (NLLH) and mean geodesic error (GE) on test data.

The results presented in Table 1 indicate that performance, in almost every case, increases substantially by including the hybrid inference step. Hence, all other experiments performed in this article are done solely using the hybrid inference where applicable (e.g., when applying the model to grid

Table 1: **Hybrid inference comparison**. Table showing mean negative log-likelihood (NLLH) and mean geodesic error (GE) for the six models specified (lower is better for both criterions), under varying amounts of synthetically generated training data with different numbers of training neurons ($N$) and different training dataset lengths ($T$).

| $N$ / $T$ | 30 / 100 | | 30 / 500 | | 30 / 1000 | | 15 / 1000 | | 45 / 1000 | |
|---|---|---|---|---|---|---|---|---|---|---|
| LVM | NLLH | GE | NLLH | GE | NLLH | GE | NLLH | GE | NLLH | GE |
| v-fae-n | 12745 | 0.57 | 8971 | 0.28 | 8781 | 0.27 | 9110 | 0.45 | 8642 | 0.27 |
| v-fae-s | 11798 | 0.64 | 9167 | 0.34 | 8949 | 0.31 | 9718 | 0.40 | 8820 | 0.31 |
| v-fae-b | 10912 | 0.50 | 9152 | 0.36 | 8830 | 0.30 | 9492 | 0.33 | 8633 | 0.26 |
| fae-n | 11975 | 0.58 | 8775 | 0.20 | 8608 | 0.17 | 9018 | 0.44 | 8415 | 0.09 |
| fae-s | 10963 | 0.42 | 8694 | 0.13 | 8657 | 0.14 | 9460 | 0.29 | 8491 | 0.13 |
| fae-b | 10040 | 0.25 | 8636 | 0.12 | 8501 | 0.12 | 9227 | 0.23 | 8267 | 0.05 |

cell data, evaluations are done in-training-sample, and without the focus on latent prediction on a test set, hence there is no need for hybrid inference at test-time).

## K  HYPERPARAMETER SEARCH

To investigate hyperparameters and their effect, we selected relevant parameters and estimated their ranges based on related literature. The relevant parameters can be seen on the left in Table 2.

To determine optimal values of hyperparameters, we relied on a random search strategy. A grid search is not feasible due to the curse of dimensionality, and a random search can also be more effective (Bergstra & Bengio, 2012). Furthermore, the random search can easily be used for discrete and continuous variables, that could be on linear or logarithmic scales. The selected scales and ranges, can be found in the middle in Table 2.

In total, we sample 500 hyperparameter configurations. Next, we train and test each configuration for 3 random seeds. We evaluate the models on the test accuracy and mean correlation. The final selected model is shown on the far right in Table 2.

Lastly, we also measure whether there is an ordinal correlation (Spearman) between the varied factors and the test LLH. The results are shown in Fig. 14. We see that learning the variance (learn_var) increases the performance on average. Also, lower learning rates and larger batch-sizes lead to better results. Lastly, also lower weighting for the KL term leads to better results.

Table 2: **Hyperparameter search.** We show our selected hyperparameters and their corresponding ranges.

| Hyperparameter | Ranges | Best |
| --- | --- | --- |
| kernel_size | choice([1, 9, 17]) | 1 |
| num_hidden | choice([16, 32, 64, 128, 256, 512]) | 128 |
| shared | choice([True, False]) | False |
| learn_coeff | choice([True, False]) | True |
| learn_mean | choice([True, False]) | False |
| learn_var | choice([True, False]) | False |
| isotropic | choice([True, False]) | False |
| num_basis | choice([1, 2, 4, 8]) | 4 |
| nonlinearity | choice([exp, softplus]) | softplus |
| batch_size | choice([1, 16, 32]) | 32 |
| batch_length | choice([64, 128]) | 64 |
| learning_rate | loguniform(1e-4, 1e-1) | 0.001003 |
| num_worse | choice([10, 50, 100]) | 10 |
| weight_kl | choice([0., loguniform(1e-9, 1e0)]) | 0.0 |
| weight_time | choice([0., loguniform(1e-9, 1e0)]) | 0.0 |
| weight_entropy | choice([0., loguniform(1e-9, 1e0)]) | 0.0 |

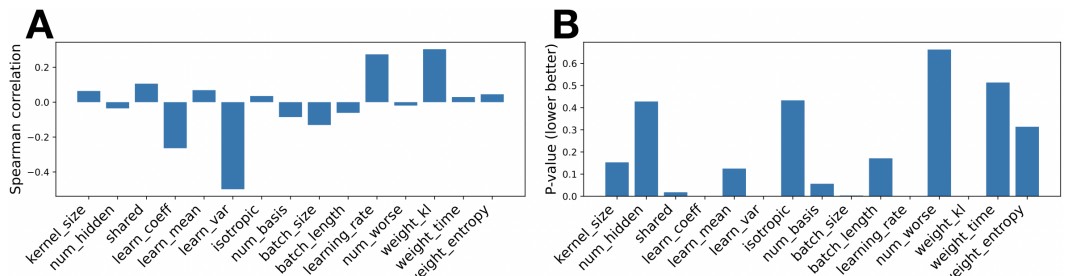

Figure 14: **Correlations with test LLH**. **A**, Spearman correlation. **B**, P-values.

