# OpenReview forum: "Understanding Neural Coding on Latent Manifolds by Sharing Features and Dividing Ensembles"
_ICLR.cc/2023/Conference — ICLR 2023 poster_

### Official Review · Reviewer_qWBb · 2022-10-24

**Confidence:** 5
**Correctness:** 4
**Technical Novelty And Significance:** 3
**Empirical Novelty And Significance:** 3
**Recommendation:** 6

**Clarity, Quality, Novelty And Reproducibility:**

The writing is mostly clear. Some need clarification.
- $x_i$ is the firing rate at the beginning of the background, and a deterministic function of the latent variable $z$. However, it seems to be stochastic according to (3).
- Why the neuron firing rate has no bias, or one of g's is constant?
- How are the ensembles determined?

The work shows a decent quality.
- Better to compare the reconstruction error for all methods. It kind of reflects how much variability is captured. Though the latent variable is perfectly encoding certain variable, it might be missing other variability in the neural activity. Since you mentioned Pei2021, why not use bps, it's available for non-Poisson likelihood too.
- Depending on the nonlinearity from latent variable to firing rate, the models might have equivalent latent variable subject to a linear/nonlinear transformation.
- vMF posterior guarantees the ring or torus. What if a non-toroidal variable is encoded? Better to evaluate it in the case of model mismatch.

The combination and use of the methods are novel.

**Strength And Weaknesses:**

Strength:
Using biologically plausible tuning curves
Spherical variational posterior captures toroidal manifold
Identifying neuron ensembles

Weaknesses:
Strong assumptions such as tuning curve and vMF posterior limit the scope of the proposed method. Most LVMs for spike trains aim at explaining most of the shared variability of the data using a low-dimensional representation. On the contrary, the demonstration of this work only shows it perfectly fits to coding toroidal behavioral variables. The vMF posterior forces the latent variable to be toroidal and mimicking Gaussian-shape tuning curve to head direction of the target neurons also forces the latent variable to be close the head direction. It is fine as   we know the toroidal head direction and the shape of tuning curves beforehand, but this demonstration makes the method seem less nontrivial.


**Summary Of The Paper:**

The authors propose a latent variable model that extracts meaningful low-dimensional representations from large scale neural recordings. The proposed method assumes Gaussian-like tuning curves to the latent variable on the neurons which share an ensemble-wise basis. The spherical variational posterior captures the toroidal manifold of variables such as head direction that is encoded in the neural activity.

**Summary Of The Review:**

The authors propose a latent variable model that extracts meaningful low-dimensional representations from large scale neural recordings. The proposed method assumes Gaussian-like tuning curves to the latent variable on the neurons which share an ensemble-wise basis. The spherical variational posterior captures the toroidal manifold of variables such as head direction that is encoded in the neural activity. Strong assumptions such as tuning curve and vMF posterior do help the inference to get latent variable desirably matches the behavior. They also limit the generalization. What more do we know from it about the data than an encoding model with the same tuning curves?

---

> ### Author Response · Authors · 2022-11-10
> **Response 1/2 to Reviewer qWBb**
>
> *Summary Of The Paper:*
>
> *The authors propose a latent variable model that extracts meaningful low-dimensional representations from large scale neural recordings. The proposed method assumes Gaussian-like tuning curves to the latent variable on the neurons which share an ensemble-wise basis. The spherical variational posterior captures the toroidal manifold of variables such as head direction that is encoded in the neural activity.*
>
> *Strength And Weaknesses:*
>
> *Strength: Using biologically plausible tuning curves Spherical variational posterior captures toroidal manifold Identifying neuron ensembles*
>
> *Weaknesses: Strong assumptions such as tuning curve and vMF posterior limit the scope of the proposed method. Most LVMs for spike trains aim at explaining most of the shared variability of the data using a low-dimensional representation. On the contrary, the demonstration of this work only shows it perfectly fits to coding toroidal behavioral variables. The vMF posterior forces the latent variable to be toroidal and mimicking Gaussian-shape tuning curve to head direction of the target neurons also forces the latent variable to be close the head direction. It is fine as we know the toroidal head direction and the shape of tuning curves beforehand, but this demonstration makes the method seem less nontrivial.*
>
> We thank the reviewer for their thorough comments and for considering our combination and use of methods as novel and of quality.
>
> To address the reviewer’s first comments:
> While it is true that the vMF posterior (which we have now correctly updated to a wrapped Normal distribution instead of a vMF in the article, thanks to the feedback from reviewer xqoS) is limited to toroidal/circular manifolds, our model is also able to discover latent variables on $\mathbb{R}^n$ (please see e.g. the Appendix for a toy example in $\mathbb{R}^2$). The model could also be adapted to consider other manifold shapes. As is, however, we expect the current model to be useful for a number of brain areas, in particular wherever one might expect to find continuous attractor dynamics – where the translation invariance of the connectivity would encourage a common shape for the tuning curves. We also note that classical work such as orientation selectivity (Hubel and Wiesel) and motor control (Georgopoulos et al. 1984) are frequently analyzed on circles, and so we argue that the circular and toroidal assumptions are not necessarily restricted to only head direction and grid cells.
>
> As for the tuning curves, both faeLVM-s and faeLVM-n utilize a sum of weighted Gaussian basis functions (i.e. a spline) to model the log tuning curves. This is a very flexible basis that does not restrict the model to only the cases where the tuning curves take a Gaussian-shape. We recognize that this might not have been stated clearly enough in the article, and we have made updates to the text accordingly.
>
> Regarding the points where there is a request for clarification:
>
> - *xi is the firing rate at the beginning of the background, and a deterministic function of the latent variable z. However, it seems to be stochastic according to (3).*
>
> The reviewer is correct regarding the ambiguous definition of $x_i$. This was a notational oversight on our side, and we have now updated the paper to clarify that $x_i$ is indeed a stochastic variable denoting neural activity, and that (the newly introduced) $\lambda_i$ denotes the instantaneous firing rate instead.
>
> - *Why the neuron firing rate has no bias, or one of g's is constant?*
>
> Our model does not currently include an explicit bias term in the equation for firing rate as we were expecting the tuning curve to largely determine the firing rate of the neurons in the situations we were considering. This would be straightforward, however, to include by incorporating e.g. an optional neuron-specific offset parameter that accounts for a non-zero background firing rate (as this would still leave the response function differentiable, and the parameter could be learned during training). We also note that faeLVM-n would already be able to incorporate an offset in firing rate, as the individual modeling of the tuning curves could account for biases in the firing rate of individual neurons.

---

> > ### Author Response · Authors · 2022-11-10
> > **Response 2/2 to Reviewer qWBb**
> >
> > - *How are the ensembles determined?*
> >
> > As for how the ensembles are learned, we wish to clarify with the reviewer whether they wonder how, in a technical sense, the participation weights $w_ij$ (which are the output of the softmax function) are learned during training, or whether they ask how the number of ensembles are determined? In the case of the former, we are happy to further expand upon the technicalities of the soft clustering procedure. In the case of the latter, the number of ensembles to infer is specified as a model parameter. In the setting where the number of ensembles would not be known, one could fit several models over a range of ensembles, then perform a comparison using cross-validation (akin to how Jensen et al. 2020 performs model selection over different manifolds) to determine the optimal number of ensembles. As mentioned in response to reviewer 3QYX, we include here preliminary results (see https://ibb.co/Yc3Xjh4) of one case with 5 circular ensembles as well as a second case with 3 tori. In both, faeLVM-b is able to accurately separate the ensembles and when testing for different numbers of ensembles, the in-sample training NLLH shows a kink at the correct number of ensembles.
> >
> > As for the reviewer’s other comments/suggestions:
> >
> > - *Better to compare the reconstruction error for all methods. It kind of reflects how much variability is captured. Though the latent variable is perfectly encoding certain variable, it might be missing other variability in the neural activity. Since you mentioned Pei2021, why not use bps, it's available for non-Poisson likelihood too.*
> >
> > We agree that the reconstruction comparison should include as many models as possible for a fair evaluation, and thank both the reviewer and reviewer xqoS for pointing this out. We have now updated the figures in question to also include the performance of the VAE (https://ibb.co/BsHkQVc, https://ibb.co/31f410D) (please also see our response to reviewer xqoS for additional analysis on the inclusion of VAE in the reconstruction error plots). We thank the reviewer for pointing out the bps measure of Pei et al. 2021, and note that in their paper, they also default to using the Poisson likelihood for the computation of the bps. While we agree that this could easily be changed to the Gaussian likelihood, we think a comparison between models using bps with a Poisson likelihood (faeLVM & VAE) and models using bps with a Gaussian likelihood (mGPLVM) would face some challenges, as we would still be comparing two different measures based on two different (although normalized) likelihood models.
> >
> > - *Depending on the nonlinearity from latent variable to firing rate, the models might have equivalent latent variable subject to a linear/nonlinear transformation.*
> >
> > We agree with the reviewer that this is a possible outcome, which could partially explain the results seen in the updated Figure 6 -- that the VAE is finding a transformation of the mouse head direction or, perhaps more likely, a combination of HD with other things. This is partially the motivation of our faeLVM and the simplified tuning curve; restricting the class of solutions to those that are most easily interpreted.
> >
> > - *vMF posterior guarantees the ring or torus. What if a non-toroidal variable is encoded? Better to evaluate it in the case of model mismatch.*
> >
> > In the case of non-circular or -toroidal variables, Equation (1) & (2) ensures that our model is still able to infer latent variables that also live on $\mathbb{R}^n$. As mentioned above, we think that $\mathbb{R}^n$ and and $\mathbb{T}^n$ would cover many neuron types (see e.g. Kriegeskorte & Wei, 2021), but we, as discussed in the Conclusion, also agree that other topologies like e.g. $SO(3)$ could be interesting avenues for further extensions of our work.

---

> > > ### Comment · Reviewer_qWBb · 2022-11-19
> > > **Response**
> > >
> > > Thank the authors for the responses. I have no further questions.

---

### Official Review · Reviewer_xqoS · 2022-10-25

**Confidence:** 3
**Correctness:** 3
**Technical Novelty And Significance:** 3
**Empirical Novelty And Significance:** 2
**Recommendation:** 6

**Clarity, Quality, Novelty And Reproducibility:**

- In equation 1 and 2, the use of subscripts and superscripts is very confusing and don’t match up with notation previously used. For instance, above equation 1 the authors state “… helps keeping the equations the same when working with multiple latents (corresponding to multiple ensembles), i.e., z = {z_1, …, z_k}” implying that k is the total number of ensembles. But in equation 1, k is being used to denote the dimension of z_j. It would help if the authors explicitly defined lower and upper limits on the product symbols.
- In section 3, the authors state that they are using a von Mises distribution as their variational posterior and prior but this seems to be at odds with section A.1 in the appendix. Specifically, from the appendix the authors seem to be using the ReLie trick from Falorsi et al., 2019. While the ReLie trick allows one to produce reparametrizable samples from S(n), this does *not* correspond to samples from the von Mises distribution (at least to my understanding).
    - **Here is a suggestion.** Use a projected normal distribution ([see Presnell et al. 1998](https://doi.org/10.1080/01621459.1998.10473768)) instead of von Mises. Unless I'm misunderstanding something, it should be fine with the reparameterization trick.
- I can’t seem to find the dimension of z is used for each of the experiments. Moreover, it isn’t also isn’t clear what latents follow a normal distribution versus a distribution on S(1). This is compounded with the fact that it also isn’t clear whether there is a subset of latents shared across the ensembles. This makes the experiments very hard to reproduce.
- In section 4.3, why isn’t the rate prediction NLL shown for the VAE case? While I understand that mGP isn’t shown because they use a Gaussian likelihood, VAEs are flexible and can be used with any likelihood distribution. This is important, especially since the VAEs seem to perform on par with faeLVM-s (with much less variability) and outperforms faeLVM-b.
- I don't fully understand how the one-hot encoding is achieved in the ensemble selection step of the network. It feels like there should be a hyperparameter controlling the temperature / sparseness of the softmax operation. Is there guidance for how users can tune up or down this sparsity?

**Strength And Weaknesses:**

The motivation behind this paper is very nice. The idea is similar to Manifold GPLVMs (Jensen et al. 2020), but in many ways the approach taken in this paper is simpler to implement and conceptualize. I really like the idea of having flexible and interpretable "deep" latent variable models for neuroscience.

The main weakness of this paper is that it uses what I would consider "easy" experimental datasets to make the case for simple tuning-curve-like decoders. For example, the grid cell data used pre-selected, non-conjunctive grid cells from MEC. This is a very special subset of the full MEC population, and it would be much more compelling if the method worked well on the raw data. At a minimum, the authors should discuss this limitation and cite [Hardcastle et al. 2017](https://doi.org/10.1016/j.neuron.2017.03.025), which characterizes the messy, conjunctive coding in the full MEC population.

In short, I think the feature sharing idea proposed by the authors can be useful on certain neural datasets, but it will be a far from universal component you'd like to build into the model. I have my doubts about whether this will become a commonly used model in the neuroscience literature (outside of Neuro-ML overlapping conference venues).

I would also like to see the paper edited for clarity, as discussed below.

**Summary Of The Paper:**

The authors propose a new latent variable model for neural data (faeLVM). The faeLVM is a generative model that combines the insights provided by tuning curves with the interpretability of low-dimensional latent variables. Specifically, faeLVM models a population of neurons are a collection of ensembles, where neurons in an ensemble are sensitive to a subset of the latent variables and share the functional form of the tuning curve, which the authors call feature sharing. For training, the authors use the VAE framework and utilize a variational posterior and optimize the ELBO. The main point of this paper, in my mind, is to make VAE-style models interpretable by making the decoder portion very simple, while still allowing for an arbitrarily complex encoder.

**Summary Of The Review:**

Overall, I like the basic idea of this paper. VAE decoders are too often treated as black boxes in scientific applications where interpretability is centrally important. I think this paper has some flaws and the empirical demonstrations are in idealized datasets. Nonetheless I am giving it the benefit of the doubt and recommending it as a borderline accept for now, pending the authors willingness to address some of the comments above.

---

> ### Author Response · Authors · 2022-11-10
> **Response 1/3 to Reviewer xqoS**
>
> *Summary Of The Paper:*
>
> *The authors propose a new latent variable model for neural data (faeLVM). The faeLVM is a generative model that combines the insights provided by tuning curves with the interpretability of low-dimensional latent variables. Specifically, faeLVM models a population of neurons are a collection of ensembles, where neurons in an ensemble are sensitive to a subset of the latent variables and share the functional form of the tuning curve, which the authors call feature sharing. For training, the authors use the VAE framework and utilize a variational posterior and optimize the ELBO. The main point of this paper, in my mind, is to make VAE-style models interpretable by making the decoder portion very simple, while still allowing for an arbitrarily complex encoder.*
>
> *Strength And Weaknesses:*
>
> *The motivation behind this paper is very nice. The idea is similar to Manifold GPLVMs (Jensen et al. 2020), but in many ways the approach taken in this paper is simpler to implement and conceptualize. I really like the idea of having flexible and interpretable "deep" latent variable models for neuroscience.*
>
> We thank the reviewer for the kind summary and appreciation of the strengths. We agree with the reviewer that model interpretability in applications aimed at broadening our understanding in e.g. the neuroscientific context is important, and we share their enthusiasm regarding “opening up” the black box, so to speak.
>
> We aim to communicate our work in as clear of a manner as possible, and therefore appreciate the reviewer’s thorough feedback on the clarity and shortcomings of our paper.
>
> *The main weakness of this paper is that it uses what I would consider "easy" experimental datasets to make the case for simple tuning-curve-like decoders. For example, the grid cell data used pre-selected, non-conjunctive grid cells from MEC. This is a very special subset of the full MEC population, and it would be much more compelling if the method worked well on the raw data. At a minimum, the authors should discuss this limitation and cite Hardcastle et al. 2017, which characterizes the messy, conjunctive coding in the full MEC population.*
>
> We understand the reviewer’s concern that the datasets might appear to be “easy”. They were chosen more as illustrations rather than to fully challenge the model. At the same time, we would argue that it is also not the easiest setting, in particular since we are dealing with 2D representations that are coding for the same underlying space (1D orthogonal representations would of course be much easier). Nonetheless, we agree that it still warrants further clarification, and we have now included a discussion of the diversity of tuning in the medial entorhinal cortex, citing Hardcastle et al., 2017, and state that there is much more to be investigated in the raw data. The changes can be found in the 1st and 3rd paragraphs of Section 4.4.
>
> *In short, I think the feature sharing idea proposed by the authors can be useful on certain neural datasets, but it will be a far from universal component you'd like to build into the model. I have my doubts about whether this will become a commonly used model in the neuroscience literature (outside of Neuro-ML overlapping conference venues).*
>
> We agree that it is not yet clear how universal the feature sharing assumption will be in neural datasets. Certainly, good candidates include any neuron population thought to be governed by continuous attractor dynamics (due to the translational invariance of the connectivity). The answer will likely become more clear soon, as the field continues to shift towards population codes, but we speculate that this assumption will not be uncommon. We also note that faeLVM can be used without feature sharing (or on just a subset of ensembles) and still benefits from the efficient inference and ensemble separation, which are both useful.

---

> > ### Author Response · Authors · 2022-11-10
> > **Response 2/3 to Reviewer xqoS**
> >
> > - *In equation 1 and 2, the use of subscripts and superscripts is very confusing and don’t match up with notation previously used. For instance, above equation 1 the authors state “… helps keeping the equations the same when working with multiple latents (corresponding to multiple ensembles), i.e., z = {z_1, …, z_k}” implying that k is the total number of ensembles. But in equation 1, k is being used to denote the dimension of z_j. It would help if the authors explicitly defined lower and upper limits on the product symbols.*
> >
> > The reviewer is absolutely correct regarding the sub/superscript mismatch in the equations, and we thank them for noticing the inconsistencies in our notation. We have updated our paper with the proper product limits, and tidied up other notational parts in Section 2 and 3.
> >
> > - *In section 3, the authors state that they are using a von Mises distribution as their variational posterior and prior but this seems to be at odds with section A.1 in the appendix. Specifically, from the appendix the authors seem to be using the ReLie trick from Falorsi et al., 2019. While the ReLie trick allows one to produce reparametrizable samples from S(n), this does not correspond to samples from the von Mises distribution (at least to my understanding).
> > Here is a suggestion. Use a projected normal distribution (see Presnell et al. 1998) instead of von Mises. Unless I'm misunderstanding something, it should be fine with the reparameterization trick.*
> >
> > We do indeed use a wrapped Normal posterior distribution as described in Appendix A and thank the reviewer for pointing out the inconsistency compared to the main methods section. We do note that the wrapped Normal and von Mises distributions are equivalent in the limit of narrow posteriors, since $e^{\kappa cos(x)}  \approx e^{-\kappa x^2/2 + \kappa x^4/4! - ...} \approx e^{-x^2/2}$ for small $x$ (to the second order), but recognize that this does not generally correspond to equivalent samples, as the reviewer points out. The wrapped Normal distribution is, however, generally easier to sample from using the reparameterization trick (as noted by the reviewer), as well as being a faster option, therefore being the preferred choice for our application. We have now updated the Methods section of the article with the correct equations for the wrapped Normal distribution.
> >
> > - *I can’t seem to find the dimension of z is used for each of the experiments. Moreover, it isn’t also isn’t clear what latents follow a normal distribution versus a distribution on S(1). This is compounded with the fact that it also isn’t clear whether there is a subset of latents shared across the ensembles. This makes the experiments very hard to reproduce.*
> >
> > For our simulated experiment in 4.1, we simulate $z \in \mathbb{S}^1$, a single latent variable where there are no ensembles. For the simulated experiment in 4.2, we simulate (respectively) two and three circular latent variables $z \in \mathbb{S}^1$, where no latents are shared across the ensembles. We concur that the explanation could have been stated more clearly, and we have updated the respective sections of the article.
> >
> > - *In section 4.3, why isn’t the rate prediction NLL shown for the VAE case? While I understand that mGP isn’t shown because they use a Gaussian likelihood, VAEs are flexible and can be used with any likelihood distribution. This is important, especially since the VAEs seem to perform on par with faeLVM-s (with much less variability) and outperforms faeLVM-b.*
> >
> > We thank both the reviewer and reviewer qWBb for commenting on this, which was indeed an oversight on our part. We have updated our figures to also include the VAE in the rate prediction showcase. We note that the VAE model has been rerun for the simulation comparisons in 4.1, as the original hyperparameter settings yielded performances that were quite a bit worse compared to the faeLVMs. In the interest of a fairer baseline comparison against the VAE, we have therefore selected a set of hyperparameters yielding better results, which you can see both in the update article and here (https://ibb.co/BsHkQVc). As for the NLLH comparison on the HD data (see here https://ibb.co/31f410D) we note that the VAE is indeed slightly better than faeLVM-n in this regard (i.e. rate prediction). This is perhaps unsurprising, since the VAE is not restricted to inferring simple tuning curves like faeLVM, allowing it to, for example, find transformations of the HD or, perhaps more likely, incorporate other variables along with the HD.

---

> > > ### Author Response · Authors · 2022-11-10
> > > **Response 3/3 to Reviewer xqoS**
> > >
> > > - *I don't fully understand how the one-hot encoding is achieved in the ensemble selection step of the network. It feels like there should be a hyperparameter controlling the temperature / sparseness of the softmax operation. Is there guidance for how users can tune up or down this sparsity?*
> > >
> > > The temperature parameter has, in the case of our model, been implicitly set to 1, and so the reviewer is correct in that there is currently no tunable option for this hyperparameter. Our observations were that the model performed well during the experiments without a temperature parameter, and in the interest of including fewer hyperparameters for potential users to deal with, we chose to omit the option. However, the addition of this additional parameter would be a simple inclusion to our model.

---

> > > > ### Comment · Reviewer_xqoS · 2022-11-25
> > > > **Keeping my score as borderline accept, it am not able to give (7 / 10)**
> > > >
> > > > I thank the authors for addressing my concerns and cleaning up notation. If I were able, I might select a score of (7 / 10), but the only options available to me are (6 / 10) or (8 / 10) so I will stick with my current score. I will be curious to see whether this model can provide future insights into more complex neural datasets with conjunctive and noisy tuning, and I thank the authors for acknowledging these future directions in their revision.

---

### Official Review · Reviewer_3QYX · 2022-10-30

**Confidence:** 3
**Correctness:** 4
**Technical Novelty And Significance:** 3
**Empirical Novelty And Significance:** 3
**Recommendation:** 6

**Clarity, Quality, Novelty And Reproducibility:**

The paper is relatively well written and the approach is clear. Some details on the different variants of the model could be elaborated upon to better understand and interpret the results.

**Strength And Weaknesses:**

Strengths:
+ The method seems to provide some good advances in both its generative capability as well as the ability to predict ensembles.

+ They introduce a new toroid task and measures of ensemble detection in this case.


Weaknesses:
- The authors compare three variants of their general approach - a shared, no-shared feature model, and a heat kernel model. While there is some discussion of how the different models provide varying benefits in different regimes (for synthetic data), the differences between the models and underlying assumptions is not entirely clear. In the experiments it seems that the shared features model doesn't give improvements across the board -- but isn't this is the core motivation of the work?

E.g., What is the difference between the non-shared model (faeLVM-n) and the geometric LVM model (mGP)? It would be helpful to have a more clear description of the ablations and other models included in the comparisons.

- In the experiments, the number of clusters is relatively low (2-3) and there isn’t much discussion about how the number of clusters is selected (or the regularization is performed to encourage soft clustering of neurons). How does the method work when there are more clusters or larger amounts of overlap between ensembles? How do you select the number of ensembles or clusters and/or regularization in the soft clustering assignment?

- The applications and datasets that are examined are somewhat limited and it's unclear how well it will work in different tasks that don’t have strong geometry and when the feature subspaces are non-overlapping. In particular, is it possible to use their method in conditions where ensembles are not as separable or more entangled?

- The set of baselines are limited. There are a number of other methods for latent variable modeling in neural data analysis, including many cited in the work and in the Neural Latents Benchmark cited in the evaluation section. It would be interesting to understand how their model compares to some of these other methods and in other datasets that don’t have clear toroid structure or where neurons are separated as cleanly as in the datasets tested here.

- For the ensemble detection method, what is the performance for the other variants of your approach? What about feature sharing? It seems this would be the natural model used for these experiments (and not just the -b model).

**Summary Of The Paper:**

This paper introduces a latent variable model for both generating and clustering neural activity. In simulated datasets, they study the performance of the model and its ability to recover underlying ensembles in a synthetic and real-world toroid task.
Overall, the paper provides an interesting model and a strong set of different evaluations of the method, across different grid cell datasets and a synthetic toroid task where they set up an ensemble detection task. However, there are some concerns about the generalization of the approach to different tasks and neural datasets.




**Summary Of The Review:**

This paper provides a new latent variable model for neural population activity that can leverage shared features across different neurons to learn population-level representations and also detect ensembles (neurons that use shared features). The model provides an interesting blend between population-level analysis and more interpretable neuron-level measures of tuning.

In empirical evaluations, the authors show that their model can provide good predictions of held out neural activity and thus serves as a good generative model, and also can recover underlying ensembles in the data. However, it is unclear how general the model is, how much they rely on simplified tasks where the subgroups are orthogonal or well separated, and how much the geometry of the underlying task matters. The relatively simple conditions in which the model is tested makes it hard to assess the generalization of the approach to different tasks or conditions, or even larger numbers of clusters.

---

> ### Author Response · Authors · 2022-11-10
> **Response 1/2 to Reviewer 3QYX**
>
> *Summary Of The Paper:*
>
> *This paper introduces a latent variable model for both generating and clustering neural activity. In simulated datasets, they study the performance of the model and its ability to recover underlying ensembles in a synthetic and real-world toroid task. Overall, the paper provides an interesting model and a strong set of different evaluations of the method, across different grid cell datasets and a synthetic toroid task where they set up an ensemble detection task. However, there are some concerns about the generalization of the approach to different tasks and neural datasets.*
>
> *Strength And Weaknesses:*
>
> *Strengths:*
> *The method seems to provide some good advances in both its generative capability as well as the ability to predict ensembles. They introduce a new toroid task and measures of ensemble detection in this case.*
>
> *Weaknesses:*
> - *The authors compare three variants of their general approach - a shared, no-shared feature model, and a heat kernel model. While there is some discussion of how the different models provide varying benefits in different regimes (for synthetic data), the differences between the models and underlying assumptions is not entirely clear. In the experiments it seems that the shared features model doesn't give improvements across the board -- but isn't this is the core motivation of the work? E.g., What is the difference between the non-shared model (faeLVM-n) and the geometric LVM model (mGP)? It would be helpful to have a more clear description of the ablations and other models included in the comparisons.*
>
> We thank the reviewer for the overall positive summary and thorough comments.
> We recognize that the distinction between our three iterations of faeLVM could have been described more clearly, and we have updated the final paragraph of Section 3 to clearly state that faeLVM-b also utilizes feature sharing. In particular, it seems that our initial lack of a clear description might have created some confusion regarding this, and so we further expand upon our explanation:
>
> While faeLVM-s shares features by inferring a common tuning shape modeled by a sum of weighted Gaussian basis functions, the heat kernel model (faeLVM-b) also shares features, namely by assuming a common fixed heat kernel shape for the log-tuning shape with learnable shared width, and a neuron-specific translation parameter. As faeLVM-b is more constrained, we would expect it to perform better with less data compared to faeLVM-s, which again performs better than the non-sharing model in the lower data regime we set out to investigate in our simulation studies (experimental length being the limitation, not the recordable number of neurons, see Stevenson & Kording, 2011). As the reviewer pointed out, one of the core motivations behind our work is indeed the benefit of feature sharing in said regime, and while we do not claim it would significantly improve models in the high data regime, we see that the models using feature sharing (faeLVM-b and faeLVM-s) do perform better than the ones without in the lower data regime.
>
> As for the difference between faeLVM-n and mGPLVM, the mGPLVM is an extension of the Wu et al. 2017-model (P-GPLVM, now LMT), which uses Gaussian Processes to model the tuning curves, in contrast to our parametric approach using a sum of weighted basis functions. Their approach also does not include the option for inferring ensemble weights, and so the mGPLVM is not capable of inference on data sets with multiple ensembles, which faeLVM-n is. In addition, since our approach utilizes the VAE-framework, we are able to perform amortized inference, which allows for much faster inference compared to the mGPLVM, and potentially also real-time inference.

---

> > ### Author Response · Authors · 2022-11-10
> > **Response 2/2 to Reviewer 3QYX**
> >
> > - *In the experiments, the number of clusters is relatively low (2-3) and there isn’t much discussion about how the number of clusters is selected (or the regularization is performed to encourage soft clustering of neurons). How does the method work when there are more clusters or larger amounts of overlap between ensembles? How do you select the number of ensembles or clusters and/or regularization in the soft clustering assignment?*
> >
> > The reason for the low number of clusters was partly due to the limitations of the comparison methods, which struggled to adequately perform the task of separating even two tori. The faeLVM model, however, is able to separate more ensembles than those showcased in our Experiments section. Although we have not performed a systematic study of this (rather it would be included in future work), we include an additional example in the Appendix of running faeLVM-b on a data set consisting of five circles (https://ibb.co/3Ythk9p), where we observe that the model is also able to accurately separate the ensembles.
> >
> > As for the selection of the number of ensembles, it is given as a hyperparameter in the model and was pre-specified in the example in the article. However, as mentioned in the reply to reviewer qWBb, one could perform model selection based on cross validation to identify the optimal number of ensembles for a particular dataset. As an illustration, we include preliminary results (see https://ibb.co/Yc3Xjh4) of the above case with 5 circles as well as a second with 3 tori (for which faeLVM-b is also able to accurately separate). We note that the in-sample train NLLH shows a kink at the correct number of ensembles. Also, while these results could be included in the current article, we would prefer to develop these ideas properly in future work focused on the model selection problem.
> >
> > - *The applications and datasets that are examined are somewhat limited and it's unclear how well it will work in different tasks that don’t have strong geometry and when the feature subspaces are non-overlapping. In particular, is it possible to use their method in conditions where ensembles are not as separable or more entangled?*
> >
> > The reviewer is correct that we have focused on applications that illustrate the method and have not yet sought to test the full extent of its applicability. In this article the focus was on the presentation of the model. We agree that future work should include additional datasets. In the case of more entangled ensembles, the model is set up so that the ensemble weights need not necessarily be one-hot, that is, a neuron could participate in more than one ensemble, which would be reflected in e.g a 0.5 - 0.5 weight split between two latents.To augment this, we would also aim to extend our framework to include multiple latents that could be shared across ensembles. For example, in the case of grid cells modules, it would be desirable to allow the neurons to participate “conjunctively” (Sargolini et al. 2006) in an additional common latent head direction circle.
> >
> > - *The set of baselines are limited. There are a number of other methods for latent variable modeling in neural data analysis, including many cited in the work and in the Neural Latents Benchmark cited in the evaluation section. It would be interesting to understand how their model compares to some of these other methods and in other datasets that don’t have clear toroid structure or where neurons are separated as cleanly as in the datasets tested here.*
> >
> > The reviewer is correct that there already exists a vast number of latent variable models for analyzing neural data. To our knowledge, this is the first one, however, to specifically work with mixtures of latent topologies (e.g., euclidean, toroidal) and parametric tuning curves. The closest model would be the mGPLVM, which considers only singular latent spaces, which we have included as a comparison model in the experiments in Fig. 4 and 6. We do agree though that a large-scale comparison between the models cited in the Neural Benchmark paper would be interesting, and a possible avenue for further work.
> >
> > - *For the ensemble detection method, what is the performance for the other variants of your approach? What about feature sharing? It seems this would be the natural model used for these experiments (and not just the -b model).*
> >
> > For the ensemble detection experiments, our intention was to showcase the performance of faeLVM versus other established methods for ensemble detection. To reduce the number of faeLVM variants in the comparison plot, we thus only included faeLVM-b (which we note does utilize feature sharing).

---

### Author Response · Authors · 2022-11-21
**Summary of Discussion Stage 1**

First of all, we would like to thank again all the reviewers, as well as the area chair, for their time and feedback on our work. We greatly appreciate the reviewers’ positive and thorough comments and questions that have further improved the rigor and quality of the paper. Specifically, to accommodate the reviewers’ remarks and responses, we have included the following experiments and clarifications in our revised manuscript:

Summary:

- Corrected ambiguous use of notation in the Background and Methods section, updated the Methods section to include the proper wrapped Gaussian distribution, as well as including clearer descriptions of our models and the experimental setups.
- Updated Figures 4 & 6 to also include the NLLH of the MLP VAE model, and added discussion around the new results [https://ibb.co/BsHkQVc], [https://ibb.co/31f410D].
- Inclusion of an ablation on a larger ensemble separation task (separating 5 circles), to show that our model is able work on a larger number of clusters [https://ibb.co/3Ythk9p].
- Preliminary results for model selection over ensembles [https://ibb.co/Yc3Xjh4].
- Clarified the setting of the ensemble detection results in the grid cell data example, citing Hardcastle et al., 2017 and discussing the diversity of tuning in MEC.

We once again thank all the reviewers for their time and comments, which has improved the clarity and quality of our work substantially. We hope that the inclusions and the changes summarized above adequately clarifies and answers the questions and comments posed by the reviewers.

---

### Decision · Program_Chairs · 2023-01-20

**Decision:**

Accept: poster

**Justification For Why Not Higher Score:**

Reviewers not overentusastic.

**Justification For Why Not Lower Score:**

The paper introduces some new features in the VAE framework that might potentially be useful in other settings as well.

**Metareview: Summary, Strengths And Weaknesses:**

This paper introduces a new latent variable modelling approach for neural spike trains that extends the normal iid latent prior to also latent distributions on a torus and extends the generative model to use tuning curves.

All reviewers liked the paper but also had a few reservations regarding the general applicability.

A good solid paper. Accept.

**Note From Pc:**

if the above contains the word "oral" or "spotlight" please see: "oral" presentation means -> notable-top-5% and "spotlight" means -> notable-top-25%. As stated in our emails, we are disassociating presentation type from AC recommendations